# Carbonyl compounds from typical combustion sources: emission characteristics, influencing factors, and their contribution to ozone formation

Yanjie Lu[1], Xinxin Feng[2], Yanli Feng[1*], Minjun Jiang[1], Yu Peng[2], Tian Chen[3,4], Yingjun Chen[2.5*]

[1] Institute of Environmental Pollution and Health, School of Environmental and Chemical Engineering, Shanghai University, Shanghai 200444, China.

[2] Shanghai Key Laboratory of Atmospheric Particle Pollution and Prevention (LAP3), Department of Environmental Science and Engineering, Fudan University, Shanghai 200438, China.

[3] Department of Environmental Health, Shanghai Municipal Center for Disease Control and Prevention, Shanghai 200336, China.

[4] State Environmental Protection Key Laboratory of the Assessment of Effects of Emerging Pollutants on Environmental and Human Health, Shanghai Municipal Center for Disease Control and Prevention, Shanghai 200336, China.

[5] Shanghai Institute of Pollution Control and Ecological Security, Shanghai 200092, China.

[1]Yanjie Lu and [2]Xinxin Feng equally contributed to this work as co-first authors.

*Correspondence to*: Yanli Feng (fengyanli@shu.edu.cn) and Yingjun Chen (yjchenfd@fudan.edu.cn)

**Abstract.** Fuel combustion is an important primary source of carbonyl compounds (CCs), yet the emission factors and influencing factors of CCs in different sources remain unclear. The emission characteristics, influencing factors, and ozone formation potentials of CCs from four combustion sources, including biomass burning (BB), residential coal combustion, on-road sources, and agricultural machineries, were investigated by field measurements. Results indicate that the emission factors of CCs from four combustion sources exhibit significant differences. Specifically, the emission factors of CCs from BB (1968.2 ± 661.2 mg/kg) are significantly higher than other sources, being an order of magnitude greater than the on-road sources (117.8 ± 78.3~576.3 ± 47.4 mg/kg). Fuel type is a key factor affecting the CCs components. BB primarily emit formaldehyde and acetaldehyde, whereas residential coal combustion exhibits a higher proportion of aromatic aldehydes and acetone. The addition of ethanol in on-road sources and biodiesel in agricultural machineries effectively promotes the formation of acetaldehyde and unsaturated aldehydes, respectively. The formation of CCs in solid and liquid fuel sources is more sensitive to combustion temperature and emission standard, respectively. Higher combustion temperature and stricter emission standard can reduce CCs emissions by 94.6% in solid fuels and by 61.3% in liquid fuel, respectively. High temperature promotes the cyclization of small molecules, supplying ample precursors for the formation of acetone and aromatic aldehydes. More attention should be paid to the ozone formation potentials of CCs from BB and agricultural machineries to alleviate the oxidizing capacity of regional atmospheres. This study provides the latest data for emission inventories and offers a scientific basis for targeted emission reduction strategies.

# 1 Introduction

Carbonyl compounds (CCs) are important intermediates in atmospheric photochemical reactions and are also significant precursors for ozone (Zhang et al., 2019; Ling et al., 2020) and secondary organic aerosols (Shen, 2013; Mellouki et al., 2015; Go et al., 2024). CCs play a crucial role in regulating the oxidizing capacity of the atmosphere and have profound effects on the environment (Duan et al., 2008; Yang et al., 2018) and human health (Krzyzanowski et al., 2008; Weisel et al., 2017). Previous studies have shown that primary emission sources in ambient air are significant contributors to CCs and significantly

influence the oxidizing capacity and photochemical reaction rates of regional atmospheres (Yang et al., 2018, 2024; Zhao et al., 2024). Tang et al. (2019) measured gaseous CCs at the junction of the Beijing-Tianjin-Hebei region and found that the main sources of small molecule CCs include vehicle emissions, solvent use, residential coal combustion (RCC) and biomass burning (BB). Zhang et al. (2020) conducted a study on the emissions of volatile compounds in the North China Plain during winter and discovered that RCC, BB, vehicle emissions, and solvent use are the primary sources of CCs emissions, consistent

with the findings of Xie et al. (2021). It is evident that vehicle emissions, BB, and RCC are typical primary emission sources of CCs. Therefore, investigating the emissions of CCs from BB, RCC, and vehicle emissions holds significant potential for controlling atmospheric oxidizing capacity.

At present, many studies have reported on the emissions of CCs from BB and RCC sources. The results indicate that the emissions of CCs are influenced by combustion conditions (temperature, oxygen concentration, stove type) and fuel

characteristics (type, composition, moisture content). For instance, Cheng et al. (2022) investigated the emission characteristics of CCs from BB, such as emission factors (EFs) and chemical composition, based on the tube-furnace. They found that combustion conditions, including temperature and oxygen concentration, as well as fuel characteristics like composition, significantly affect the EFs and composition of CCs. Peng et al. (2023) explored the emission characteristics of CCs from poplar with varying moisture content and discovered that altering the moisture content can effectively reduce the formation of

CCs. He et al. (2024) conducted field measurements of pollutant emissions from combustion of animal dung and coal in the Tibetan Plateau and found that the differences in EFs can be explained by combustion conditions and fuel type. Similarly, Liu et al. (2022) conducted experiments with 19 types of fuel and tube-furnace at low and high temperature, finding that both fuel type and stove type influence the emission characteristics and composition of CCs. Furthermore, the $V_{daf}$ (the volatile matter content on a dry and ash-free basis) significantly affects the emission of CCs during RCC. For example, Feng et al. (2010)

found that the concentrations of CCs and VOCs increase and then decrease with the maturity of RCC. However, current research lacks a systematic analysis of the impact of the different $V_{daf}$ of coal samples on the emission characteristics of CCs. There is an urgent need to reveal the influence of $V_{daf}$ on the emission characteristics of CCs from RCC. Additionally, the effects of different fuel type and combustion system on the emission characteristics of CCs from BB and RCC sources need to be assessed, along with their contributions to atmospheric oxidizing capacity.

Numerous researchers have employed various methods to determine the emission characteristics of CCs from on-road sources, such as chassis dynamometer tests (Nelson et al., 2008; Guo et al., 2011; Karavalakis et al., 2011), tunnel experiments (Ho et

al., 2007; Hung-Lung et al., 2007; Wu et al., 2021), and portable emission measurement system (Yao et al., 2015; Cao et al., 2020; Wang et al., 2024). Studies have found that different emission standard, fuel type, and vehicle speed all influence the emission characteristics and composition of CCs in vehicle exhaust. For instance, Liu et al. (2024) conducted emission tests on 15 light-duty gasoline vehicles using a chassis dynamometer and found that the progressive tightening of emission standard can significantly reduce CCs emissions, and the use of ethanol-blended gasoline can decrease the emissions of volatile organic compounds. Song et al. (2010) measured CCs emissions from diesel engines and found that both fuels typically exhibit higher CCs emissions at high speed compared to low and medium speed. Ethanol-gasoline is characterized by its high combustion efficiency, high octane rating, and renewable nature (Costagliola et al., 2013; Zaharin et al., 2018). So it has been widely used as a partial substitute for gasoline and has been promoted and applied. Therefore, with the progressive tightening of emission standard and the improvement of fuel type and quality in China, it is particularly necessary to re-evaluate the impact and mechanisms of different emission standard and fuel type (such as ethanol-gasoline) on the emission characteristics of CCs from on-road sources.

Agricultural machineries (AMs), as an important component of non-road mobile sources, plays a significant role in seasonal severe pollution events through its emissions of CCs. Studies have found that different emission standard and fuel type significantly affect the emissions of CCs from AMs. For example, Cao et al. (2020) studied the impact of biodiesel on CCs emissions using the portable emission measurement system and found that stricter emission standard can significantly reduce the EFs of CCs ($EF_{CCs}$), while the use of biodiesel has different effects on CCs emissions from different vehicles. Yu et al. (2023) tested the emissions of CCs from 20 AMs with different emission standard using PEMS and found that stricter emission standard significantly reduce CCs emissions. In recent years, the Chinese government has actively promoted the development of the biodiesel industry, including applications in vehicles and ships. Studies have found that the use of biodiesel significantly reduces emissions of conventional pollutants such as CO and PM (Singh et al., 2016; Lou et al., 2022), but its impact on CCs emissions is still not entirely clear. Therefore, it is necessary to assess the impact of biodiesel on CCs emissions, as well as the effects of different blends of diesel on the emission characteristics of CCs, and further explore their contributions to atmospheric oxidizing capacity.

To effectively control the emissions of CCs, it is necessary not only to supplement and update data from various combustion sources but also to conduct more detailed studies on the influencing factors of CCs emissions from BB and RCC, on-road sources, and non-road AMs sources. In this study, the emission characteristics and the key influencing factors of CCs were explored based on real-world measurements from four typical combustion sources. For BB and RCC, experiments were carried out in the real-stove and tube-furnace simulation combustion to investigate the effects of fuel type (particularly the Vdaf of coal) and combustion temperature (500℃, 800℃) on CCs emissions; as for on-road source, the urban closed-loop road measurements for gasoline and diesel vehicles were conducted to explore the effects of fuel type (such as ethanol-gasoline), driving speed (low/medium/high speed), and emission standard (China V and China VI) on CCs emissions; and concerning the non-road AMs sources, the effects of emission standard and fuel type (such as biodiesel) on CCs emissions were conducted based on field vehicle measurements. Finally, the contributions of the four typical sources to atmospheric oxidizing capacity

were estimated by the ozone formation potentials (OFPs). This study will help deepen our understanding of CCs emissions from different combustion sources, provide the latest data for improving emission inventories, and offer a scientific basis for future air quality simulations and pollution control strategies.

## 2 Materials and Methods

### 2.1 Tested fuels and vehicles

Twelve common fuels were selected, including three types of straw (rice straw, wheat straw, and corn straw, which are widely cultivated in southern and northern China), three types of wood (pine, poplar, and willow, representing softwood and hardwood), and six types of coal with different $V_{daf}$ (LL coal, GJ coal, DT coal, SH coal, NM coal, and PX coal) (Table S1). These fuels are typically used for cooking, heating, or burning activities in rural areas of China. In laboratory conditions, the same fuels (2g) were used as controls, with combustion temperatures set at 500℃ and 800℃ to simulate ordinary stoves and high-efficiency energy-saving stoves commonly used in daily life. A total of 43 samples from actual stoves and 94 samples from laboratory tube-furnace simulations were obtained for BB and RCC sources.

On urban closed-loop roads, the emission characteristics of CCs in the exhaust of gasoline vehicles (GVs) and diesel vehicles (DVs) were tested, examining the effects of fuel type, vehicle speeds, and emission standard on their emissions. Passenger and commercial vehicles meeting the China V and China VI emission standards (2-4 vehicles per category) were selected to assess the emission characteristics of regular gasoline, regular diesel, and ethanol-blended gasoline (Ethanol gasoline vehicles, E-GVs) under different driving speeds, including low speed (30 km/h), medium speed (60-90 km/h), and high speed (120 km/h) (Table S2). Field vehicle measurements were conducted on non-road AMs in farmland, selecting tractors and harvesters meeting the China II and China III emission standards as experimental machinery to evaluate the emission characteristics of CCs in the exhaust of regular commercial diesel (B0) and diesel/biodiesel blended fuels (Table S3). The biodiesel was produced from waste cooking oil and mixed with commercial fossil diesel to create two blended fuels with different ratios (B5, B20). A total of 67 samples from on-road sources and 69 samples from AMs were obtained for mobile source sampling.

### 2.2 Sample collection and analysis

Emissions from typical combustion sources, including flue gas or exhaust, were collected using a laboratory-made sampling dilution system, which comprises a dilution system and a sampling system. The detailed description of laboratory-made sampling dilution system was presented in previous studies(Cheng et al., 2022; Liu et al., 2022; Peng et al., 2023; He et al., 2023). A schematic diagram of the sampling system is presented in Figure 1. Briefly, the clean air was introduced into dilution system via a fan, where it is mixed with the flue gas to achieve the purpose of diluting the flue gas. And the concentration of $CO_2$ and CO in the flue gas was monitored in real-time using a flue gas analyzer (Photon-II, Madur, Italy) to control the dilution ratio of the flue gas. The collection of CCs from the flue gas was accomplished using an automatic CCs sampler equipped with 2, 4-dinitrophenylhydrazine (DNPH) coated cartridges. CCs can react with DNPH to form stable hydrazone derivatives(Tang

et al., 2003). The sampling flow rate was 0.5 L/min and DNPH cartridges were placed behind quartz filters to collect gaseous CCs. After collection, the samples were immediately refrigerated at -4℃ until analysis. The DNPH sampling tubes were placed in a vacuum glove box, and 2-3 mL of acetonitrile was used to elute the target compounds, which were then transferred to

vials. The samples were quantitatively analyzed using high-performance liquid chromatography (HPLC) (1260; Agilent, USA) equipped with a UV detector and an Agilent TM C18 reverse-phase chromatographic column (5.0 μm, 250 mm × 4.6 mm). The mobile phase consisted of a mixture of water, acetonitrile, and tetrahydrofuran in a specific ratio to ensure optimal separation efficiency. A total of 20 CCs were analyzed, including formaldehyde (FA), acetaldehyde (ALD), acetone (ACE), unsaturated aldehydes (UA) (acrolein, crotonaldehyde), aromatic aldehydes (AA) (benzaldehyde, m/p-methylbenzaldehyde,

o-methylbenzaldehyde, dimethylbenzaldehyde), and other aldehydes and ketones (Other CCs) (propanal, butaldehyde, cyclohexanone, isopentanal, pentanal, heptanal, octanal, nonanal, decanal). QA/QC details are provided in the supplementary materials.

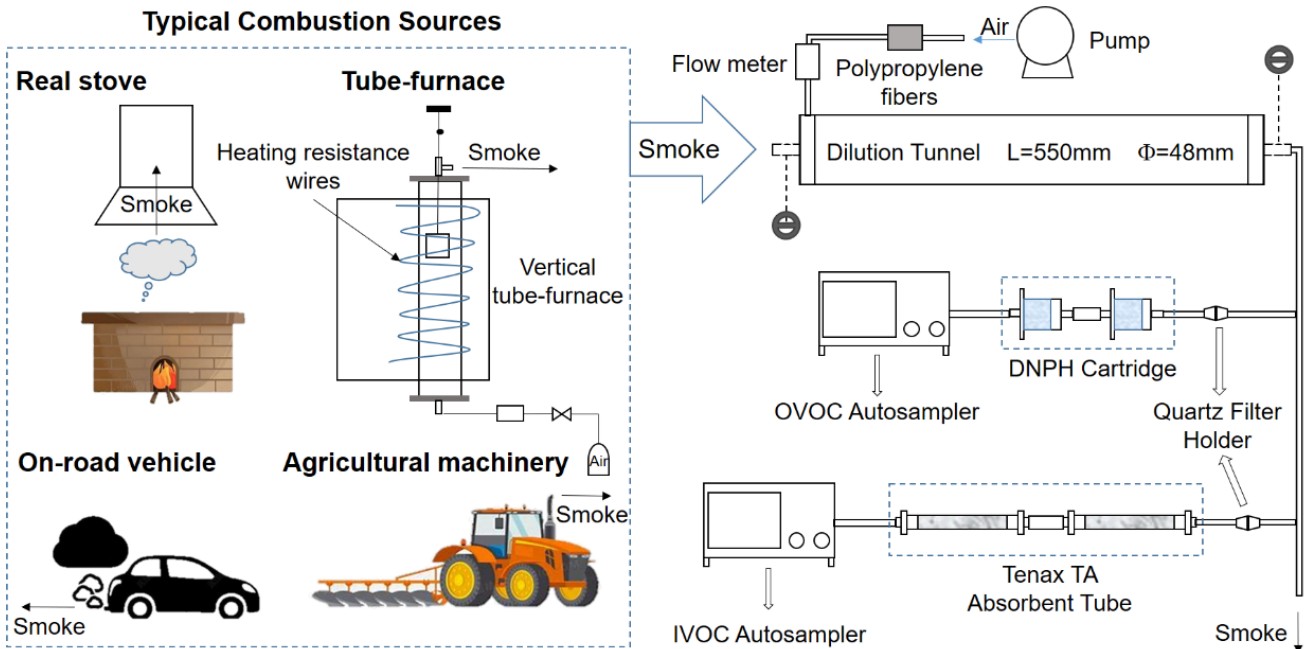

**Figure 1: The sampling system diagram**

**2.3 Calculation of the emission factors**

In this study, the EF$_{CCs}$, based on the carbon balance, were calculated by Eq. (1)-(6)(Qian et al., 2021; Liu et al., 2022; Zhang et al., 2024). It is assumed that during combustion, all carbon in the fuel (excluding carbon in the ash) is primarily converted into CO, CO$_2$, and particulate carbon (OC, EC), i.e.:

$$m_{c-fuel} - m_{c-ash} = m_{c-CO_2} + m_{c-CO} + m_{c-OC} + m_{c-EC} \tag{1}$$

Incomplete combustion factor:

$$K = \frac{m_{c-CO} + m_{c-OC} + m_{c-EC}}{m_{CO_2}} \tag{2}$$

EFs of carbon in the form of $CO_2$:

$$EF_{CO_2} = \frac{m_{c-fuel} - m_{c-ash}}{(1+K)M} \tag{3}$$

When the mass of carbon in the ash is significantly lower than the mass of carbon in the fuel:

$$m_{c-fuel} - m_{c-ash} = Mk_{c-fuel} \tag{4}$$

where, $m_{c-fuel}$ and $m_{c-ash}$ represent the carbon mass in the fuel and ash, respectively. $m_{c-CO_2}$ and $m_{c-CO}$ denote the carbon mass in $CO_2$ and CO, respectively. $m_{c-OC}$ and $m_{c-EC}$ are the masses of organic carbon and elemental carbon in particulate matter, respectively. $M$ is the mass of residential solid fuel (kg), and $k_{c-fuel}$ is the carbon content of the residential solid fuel determined by elemental analysis.

$$EF_{CO_2} = EF_{C-CO_2} f_{CO_2} = \frac{(m_{c-fuel} - m_{c-ash}) f_{CO_2}}{(1+K)M} = \frac{k_{c-fuel} f_{CO_2}}{(1+K)} \tag{5}$$

$$EF_i = \frac{C_i}{C_{CO_2}} EF_{CO_2} \tag{6}$$

where $f_{CO_2}$ is the conversion factor, taken as 3.67. $C_i$ and $C_{CO_2}$ are the mass concentrations of target compound i and $CO_2$, respectively (μg/L), and $EF_{CO_2}$ is the emission factor of $CO_2$.

The modified combustion efficiency (MCE) is a widely accepted evaluation parameter in biomass combustion research(McMeeking et al., 2009; Shen et al., 2010; Li et al., 2016), which can be used to represent the combustion condition:

$$MCE = \frac{\Delta CO_2}{(\Delta CO_2 + \Delta CO)} \tag{7}$$

where $CO_2$ and CO are the concentrations of $CO_2$ and CO emitted from the fuel combustion.

## 2.4 Oxidizing capacity assessment calculation

The OFPs of CCs can be predicted based on the maximum incremental reactivity (MIR) values of specific species, and the specific Eq. (7) is as follows(Yu et al., 2023; Zhang et al., 2024).

$$OFP = \Sigma(MIR_i \times EF_i) \tag{7}$$

where $OFP_i$ is the ozone formation potential of a specific species i in CCs (g $O_3$)/(kg-fuel); $MIR_i$ is the maximum incremental reactivity of species i (Table S4) (Zhang et al., 2021), and $EF_i$ is the emission factor of CCs species i (g/(kg-fuel).

Uncertainty analysis for OFP calculations is provided in Supplement Text S1.

**2.5 Statistical significance testing**

The t-test is a widely utilized statistical method for comparing mean differences between samples. In this study, data analysis was performed using SPSS 26.0. To evaluate the significance of mean differences between the two groups, the normality of the data was first assessed via the Kolmogorov-Smirnov test. Upon confirmation of normality, the homogeneity of variances was subsequently examined using Levene's test. If variance homogeneity was satisfied, an independent samples t-test was applied for intergroup comparisons; otherwise, Welch's corrected t-test was employed. Statistical significance was determined at a 95% confidence interval (CI), with a two-tailed significance threshold set at $\alpha = 0.05$. Results were reported as t-values, degrees of freedom (df), and p-values. A statistically significant difference was defined as $p < 0.05$. The analytical procedure strictly adhered to the classical t-test assumptions (independence, normality, and homogeneity of variance) to ensure methodological rigor.

**3 Results and discussion**

**3.1 Emission characteristics and influencing factors of CCs from BB and RCC sources**

For real stoves, the EF of CCs ($EF_{CCs}$) from BB (1676.4 ± 989.5 mg/kg) is found to be significantly higher than that from RCC (287.9 ± 79.2 mg/kg) (t-test, $p > 0.05$) (Fig 2). The difference is attributed to the high emission of CCs being closely related to the high oxygen content of the fuel, such as the oxygen content of biomass being much higher than that of residential coal. Fig 2 presents the $EF_{CCs}$ emitted from the combustion of two types of biomass (straw and wood) and residential coal. The $EF_{CCs}$ from BB and RCC are 1676.3 ± 989.5 mg/kg and 287.9 ± 79.2 mg/kg, respectively. The $EF_{CCs}$ from BB is approximately eight times that of RCC. Among these, the $EF_{CCs}$ from straw combustion (2384.1 ± 1515.0 mg/kg) is about 2.5 times that of wood combustion (968.6 ± 464.0 mg/kg), which may be attributed to differences in fuel structure and moisture content (Cheng et al., 2022; Peng et al., 2023).

The composition of CCs is significantly influenced by fuel type. Formaldehyde and acetaldehyde are the most abundant CCs in all three types of fuel, accounting for 71.3% (straw combustion), 74.8% (wood combustion), and 67.0% (RCC) of the total CCs. Although the $EF_{CCs}$ from BB is much higher than that from RCC, the proportion of acetone in CCs emitted from RCC is higher than that from BB, representing 10.3% of the total CCs, while straw combustion and wood combustion account for 5.8% and 6.9%, respectively. There are significant differences in the proportion of unsaturated aldehydes among different fuels, with the proportion in straw combustion being 1.7 times that of wood combustion and 2.9 times that of RCC. For aromatic aldehydes, the proportion in RCC is the highest (10.6%), while straw combustion and wood combustion account for 2.9% and 4.4%, respectively. The reason for this phenomenon is that the chemical structure of residential coal contains a higher content of aromatic compounds than biomass, which decompose during combustion, providing a large number of precursors for the formation of aromatic aldehydes. Additionally, the large error bars in each type of fuel indicate variability in emissions within

the fuel, which may be related to combustion conditions. Therefore, this study conducted simulation experiments in the tube-furnace to further explore the key influencing factors of CCs emissions from BB and RCC sources.

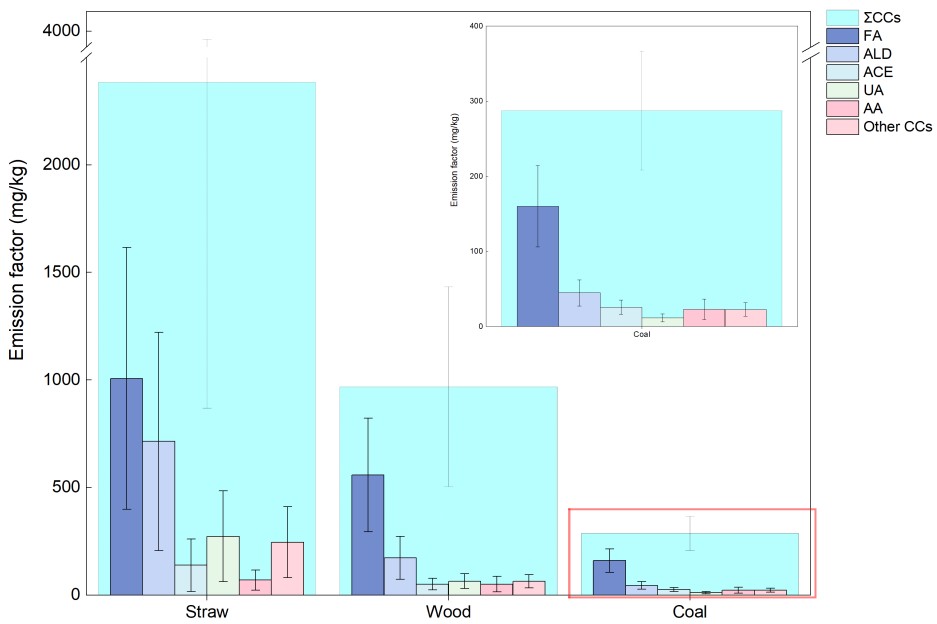

**Figure 2: EF$_{CCs}$(mg/kg)in real stoves from BB and RCC sources**

Fuel type significantly influences CCs formation across straw combustion, wood combustion, and RCC. For straw, the CCs generated from the combustion of southern straw (rice straw) are higher than those from northern straw. The EF$_{CCs}$ from rice straw combustion (3865.6 ± 558 mg/kg) is significantly higher than that of corn (2829.8 ± 1771.8 mg/kg) and wheat (1772.6

± 847.2 mg/kg), indicating that the CCs generated from southern straw combustion are 1.4-2.2 times higher than those from northern straw combustion. This phenomenon may be attributed to differences in the content of biomass components (cellulose, hemicellulose, and lignin) and combustion efficiency. Cheng et al. (2022) found that among the three biomass components, cellulose combustion generates the highest amount of CCs, followed by hemicellulose, with lignin producing the least. Meanwhile, Zhao et al. (2019) discovered that the holocellulose content (~56.3%) of rice straw is higher than that of corn and

wheat. Additionally, in this study, the MCE of rice (90.9%) is slightly lower than that of corn and wheat (MCE: 92.0–92.8%). As for wood, the CCs generated from the combustion of softwood (pine) are higher than those from hardwood. The EF$_{CCs}$ from pine combustion (1415.3 ± 431.8 mg/kg) is significantly higher than that of poplar (1020.3 ± 249.1 mg/kg) and willow (905.5 ± 109.6 mg/kg), indicating that the CCs generated from softwood combustion are 1.4-1.6 times higher than those from hardwood. This phenomenon may be due to differences in combustion efficiency. In this study, the MCE of pine (94.0%) is

significantly lower than that of poplar and willow (MCE:95.6–96.0%). Studies have shown that incomplete combustion of

fuels is more likely to generate CCs. Furthermore, for residential coal, the $V_{daf}$ also affects the emission of CCs from RCC. As the $V_{daf}$ increases, the $EF_{CCs}$ first decreases (212.1 ± 99.2 mg/kg) and then reaches a maximum value (372.9 ± 53.7 mg/kg) when the $V_{daf}$ is around 30%. According to the bell-shaped distribution theory by Chen et al. (2009), coal samples around 30% have special chemical characteristics, and these samples produce a large amount of coal tar during combustion, which further

increases the formation of CCs (Du and Li, 2022). In contrast, for coal types with the $V_{daf}$ between 20% and 30%, the $EF_{CCs}$ decreases by 43.1%.

It should be mentioned that temperature is a key factor influencing the formation of CCs from BB and RCC sources. Fig. 3a and 3c show that the $EF_{CCs}$ from BB (straw and wood) and RCC under different influencing factors (temperature, $O_2$ concentration, moisture content, and $V_{daf}$). High temperature significantly reduce the formation of CCs from BB and RCC

sources, with reductions of 92.3%, 96.8%, and 51.1% for straw, wood, and residential coal, respectively. The reduction effect of high temperature on CCs from BB is particularly pronounced. Compared to combustion temperature, the moisture content also has a certain impact on the formation of CCs during BB. For instance, when the $V_{daf}$ of fuels is close to 10%, the $EF_{CCs}$ is the lowest (832.0 ± 160.2 mg/kg). As the moisture content increases, the $EF_{CCs}$ gradually rises, reaching a peak (2008.7 ± 397.1 mg/kg) when the moisture content approaches 30%, which is about 2.4 times the minimum value (Peng et al., 2023).

Therefore, selecting the optimal moisture content (10%) can achieve a maximum reduction effect of 58.6%. Similarly, the increase in $O_2$ concentration significantly reduces the formation of CCs. At the $O_2$ concentration of 21%, the $EF_{CCs}$ for straw and wood are 4613.0 mg/kg and 6545.6 mg/kg, respectively. Compared to the $O_2$ concentration of 10.5%, the average $EF_{CCs}$ from straw combustion and wood combustion decrease by 33.2% and 36.8%, respectively. Therefore, increasing combustion temperature and $O_2$ concentration, selecting biomass fuels with the $V_{daf}$ of 10%, or choosing coal samples with medium to low

$V_{daf}$ (around 26%) can effectively reduce the formation of CCs.

Different CCs species exhibit varying sensitivities to combustion temperature. Specifically, for BB and RCC sources, high temperature facilitate the degradation of formaldehyde and acetaldehyde while promoting the formation of acetone and aromatic aldehydes. As shown in Fig 3b and 3d, the EFs of formaldehyde, acetaldehyde, and acetone are significantly higher than those of other CCs, accounting for about 62.8% of the total CCs. This is consistent with the results of Schauer et al. (2001)

and Liu et al.(2022). With the increase of combustion temperature, the proportions of formaldehyde and acetaldehyde in straw, wood, and residential coal decrease by 34.6%, 37.7%, and 30.7%, respectively. In contrast, at higher temperature, the proportions of acetone and aromatic aldehydes in the three types from BB and RCC sources increase by 19.8%, 19.6%, and 7.2%, respectively, compared to those at lower temperature. It should be mentioned that small molecules, including formaldehyde and acetaldehyde, are more sensitive to combustion temperature compared to larger molecules, due to their

tendency to polymerize or cyclize under high-temperature conditions (Peng et al., 2023). The mechanisms of acetone and aromatic aldehyde formation in the flue gas from BB and RCC at high temperature are significantly different. For BB, the formation of acetone and aromatic aldehydes is closely related to the pyrolysis of lignin (Cheng et al., 2022). For example, high temperature promotes the dehydration and isomerization of 1,2-propanediol, generated from the cleavage of alcoholic hydroxyl groups of lignin, to form acetone (Caballero et al., 1997). Similarly, high temperature facilitates the formation of

aromatic aldehydes such as benzaldehyde from the cleavage of aromatic ring side chains of lignin (Wang et al., 2017). In contrast to BB, the increase in the proportion of acetone and aromatic aldehydes in RCC is attributed to secondary gas-phase reactions at high temperature. The conjugated bonds in the aromatic ring structure of coal break at high temperature, providing more precursors for the formation of acetone (Miura, 2000). At high temperature, aromatic hydrocarbons generated from coal combustion undergo secondary reactions and oxidation to form aromatic aldehydes such as benzaldehyde (Wang et al., 2017).

This mechanism is further supported by the pattern of aromatic hydrocarbons observed at high temperature. The pattern of aromatic hydrocarbons were reported by Qian et al. (2021) at high temperature further verifies this mechanism. Unsaturated aldehydes are insensitive to temperature changes, with their proportions fluctuating between 4.3% and 6.5% in the three types from BB and RCC sources.

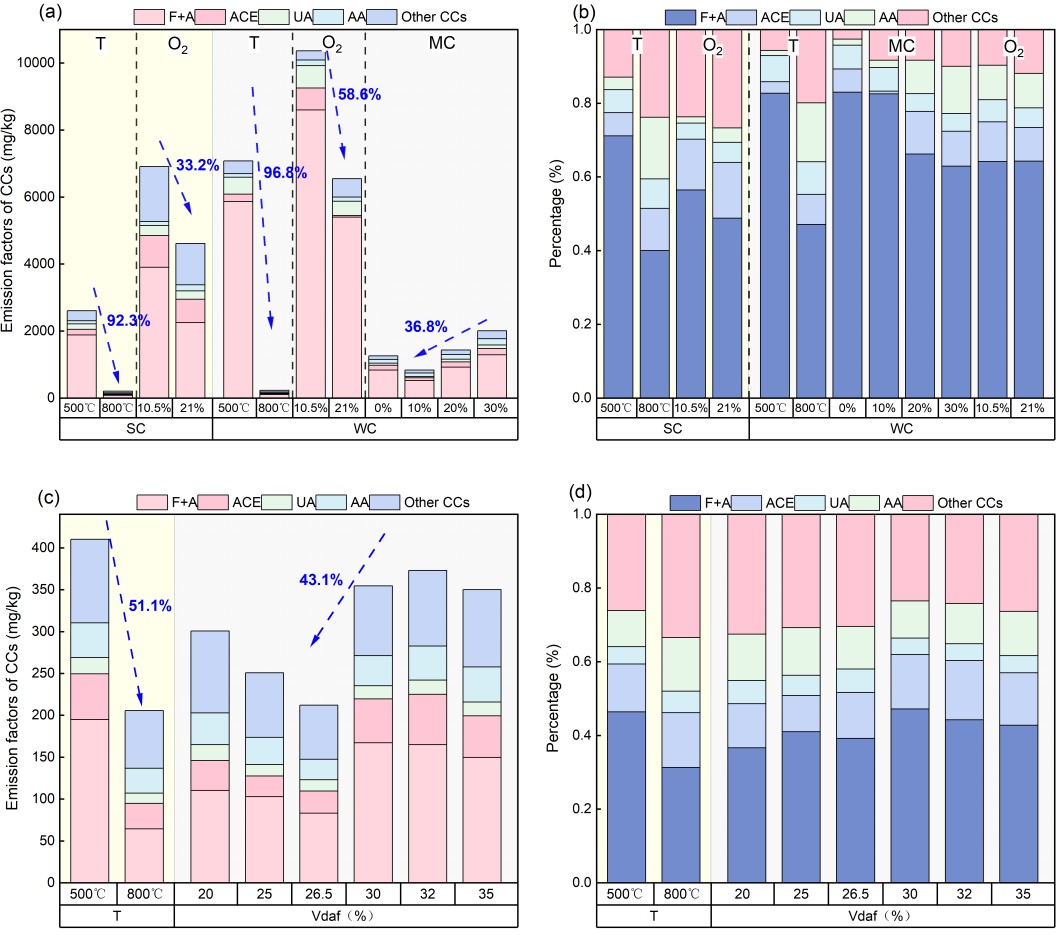

**Figure 3: Differences in CCs emission characteristics and composition from BB and RCC sources under different influencing factors; T, Temperature; O₂, Oxygen concentration; F+A, Formaldehyde+ Acetaldehyde; SC, Straw Combustion; WC, Wood Combustion.**

## 3.2 Emission Characteristics and Influencing Factors of CCs from On-road Sources

The formation of CCs from on-road vehicles is significantly influenced by fuel type and emission standard. As shown in Fig. 4a and 4b, the $EF_{CCs}$ exhibits that DVs emit more CCs than GVs, with the $EF_{CCs}$ of 214.1 ± 34.1 mg/kg for DVs and 117.8 ± 78.3 mg/kg for GVs. This result has also been frequently observed in previous findings (Ban-Weiss et al., 2008; Martinet et al., 2017; Wang et al., 2024), indicating that fuel type is a critical factor in the formation of CCs from on-road sources, achieving the 45.0% reduction. The higher emission from diesel vehicles may be related to their combustion process, where excess air is present in the combustion chamber (i.e., the overall fuel-lean condition), leading to higher oxygen content and more oxidation processes during combustion (Pang et al., 2008; Gentner et al., 2017). Additionally, the use of ethanol-blended gasoline significantly reduces formaldehyde emissions but increases the EFs of acetaldehyde, changing the ratio of formaldehyde to acetaldehyde from 5.6:1 to 1:1. The EFs of acetaldehyde in E-GVs is 2.8 times that of GVs (Magnusson et al., 2002). When 10% ethanol is added, the change in ethanol content has a minimal impact on the total emissions of CCs. This phenomenon is consistent with existing research findings, such as those by Karavalakis et al. (2014, 2015)and Yang et al. (2019), who also found that the use of ethanol gasoline significantly affects the formation of acetaldehyde, while having a minor impact on total CCs emissions. Similarly, the study revealed that China VI emission standard E-GVs exhibit an acetaldehyde-to-formaldehyde (A/F) ratio of approximately 0.38 in CCs emissions, which is about 2.5 times that of conventional GVs meeting the same emission standard. This demonstrates that ethanol gasoline application significantly enhances acetaldehyde emissions. In contrast, this A/F ratio for China V E-GVs is 1.80, suggesting that higher emission standard can effectively reduce CCs emissions while also decreasing acetaldehyde generation. The reason for this phenomenon may be that, with the progressive tightening of emission standard, exhaust treatment technologies are improved, and emission requirements become stricter, leading to further oxidation of harmful substances such as acetaldehyde into formaldehyde. Additionally, when emission standard is tightened, the average $EF_{CCs}$ for DVs and GVs decrease by 61.3% and 23.9%, respectively, indicating that stricter emission standard have a more significant reduction effect on DVs emissions. This also suggests that the impact of emission standard on CCs emissions from GVs is much lower than that on diesel vehicles. Vehicle speed is equally important for CCs formation. Data analysis reveals that for GVs, DVs, and E-GVs, the distribution of $EF_{CCs}$ follows the pattern of low speed > high speed > medium speed (Figure 3). That is, at medium speeds, CCs emissions reach their minimum value (71.2 ± 21.1 mg/kg to 177.9 ± 22.8 mg/kg), with a maximum reduction of 55.9% in CCs emissions. This conclusion is consistent with the findings of Liu et al.(2024), and the likely reason is that combustion efficiency is highest at medium speed, effectively reducing the formation of CCs. This phenomenon indicates that the impact of vehicle speed on emissions is nonlinear, and an optimal driving speed can enhance combustion efficiency, thereby effectively reducing CCs emissions.

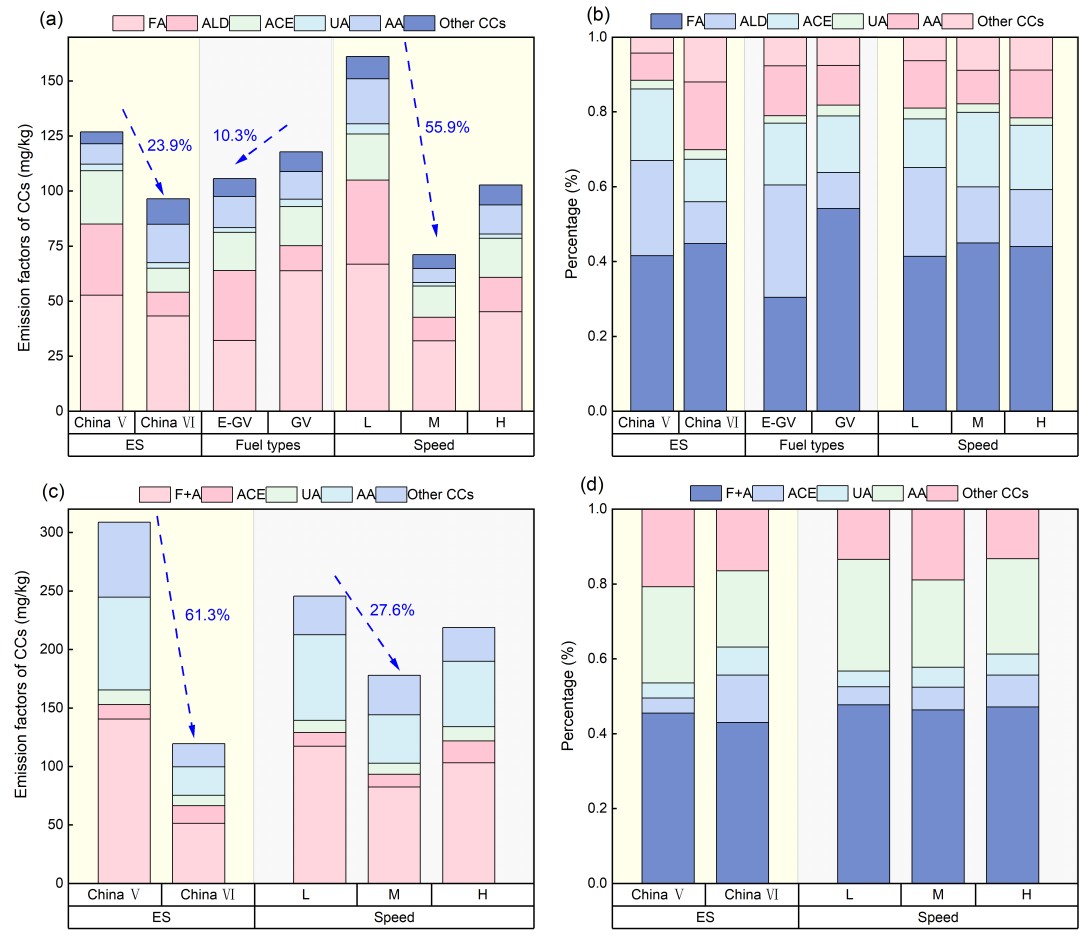

**Figure 4: Differences in CCs emission characteristics and composition of gasoline vehicles(a、c) and diesel vehicles(b、d); ethanol-gasoline vehicles; ES, Emission standard; L, Low speed; M, Medium speed; H, High speed; FA, Formaldehyde; ALD, Acetaldehyde.**

Additionally, the type and composition of CCs were directly determined by fuel type (Figure 4c, 4d). GVs primarily emit small molecule CCs, such as formaldehyde, acetaldehyde, and acetone, accounting for about 78.9% of total CCs. In contrast, DVs emit more large-molecule aldehyde and ketone compounds, such as aromatic aldehydes (46.1%) (Gentner et al., 2012; Erickson et al., 2014; Yue et al., 2015), highlighting the importance of reducing emissions of heavier aromatic aldehyde CCs in DVs. This difference mainly stems from the compositional differences in the fuels. For example, diesel contains a higher proportion of medium to high carbon hydrocarbons ($C_9$-$C_{23}$), especially polycyclic aromatic hydrocarbons, which are more likely to oxidize to form aromatic aldehyde compounds. Gasoline, in contrast, primarily consists of lower carbon hydrocarbons ($C_4$-$C_{12}$) (Gentner et al., 2017; Sorokina et al., 2021), resulting in the formation of more small molecule CCs. An inverse trend was observed for the effects of emission standard on the composition of CCs from different fuel type. For instance, when emission

standard is tightened, the proportions of formaldehyde and acetaldehyde decrease for both GVs and DV, while the proportions of acetone and aromatic aldehydes show the opposite trend. The real reason for the difference in result was mainly attributed to the different exhaust treatment devices in the two types of vehicles, leading to varying removal efficiencies for the same pollutants (Russell and Epling, 2011). The impact of vehicle speed on the composition of CCs is not significant.

### 3.3 Emission characteristics and influencing factors of CCs from non-road sources

In the emissions of CCs from AMs, emission standard is considered as a key factor influencing emission characteristics. Figure 5a illustrates the $EF_{CCs}$ and composition from AMs under different emission standard and fuel type. Similar to on-road vehicles, emission standard significantly affect the formation of CCs from AMs. Specifically, the $EF_{CCs}$ from China III standard AMs is significantly reduced compared to that from China II standard machinery (by 57.0%). This result indicates that raising emission standard can significantly decrease the formation of CCs from AMs, especially for older China II standard machinery, where upgrading emission standard offers substantial potential for emission reduction. Additionally, the study found that the $EF_{CCs}$ increased from 576.3 mg/kg for B0 to 785.9 mg/kg for B20, representing a 26.7% increase in CCs emissions (Figure 5a). This trend is consistent with the conclusions drawn by Karavalakis et al. (2011) and Prokopowicz et al. (2015) in existing literature, who suggested that the ester-based oxygen structure of biodiesel may be a significant factor contributing to the increase in CCs emissions. The higher oxygen content in biodiesel facilitates the formation of more carbon oxides during combustion, thereby increasing CCs emissions. Moreover, biodiesel may also affect the calorific value and density of the blended fuel, leading to decreased combustion efficiency (Guo et al., 2023), which further promotes the formation of CCs. With the increase in the proportion of biodiesel blending, some specific species in CCs change significantly. For example, the EFs of acrolein in B20 increased by 40.7% compared to B0, attributed to the oxidation of residual glycerol, saturated fatty acids, and unsaturated fatty acids in biodiesel (Graboski and McCormick, 1998). In contrast, since biodiesel does not contain aromatic hydrocarbons, the addition of 20% biodiesel reduced the EFs of aromatic aldehydes by 21.5% (Karavalakis et al., 2011). Additionally, the $EF_{CCs}$ from AMs using diesel (576.3 ± 47.4 mg/kg) is 2.7 times that of on-road DVs (214.1 ± 85.3 mg/kg). The likely reason for this phenomenon is that emission standard (exhaust treatment technologies) for on-road diesel vehicles can reduce the CCs emissions of diesel (Sharp et al., 2000; Russell and Epling, 2011; Wang et al., 2022). The importance of updating emission standard for both on-road and non-road sources of CCs emissions was reaffirmed. In summary, although increasing the proportion of biodiesel blending can reduce emissions of some conventional pollutants such as CO and PM(Chien et al., 2009; Karavalakis et al., 2010; Lin et al., 2011), it may also have adverse effects on CCs emissions. This necessitates a comprehensive consideration of its emission reduction benefits and potential increases in emissions.

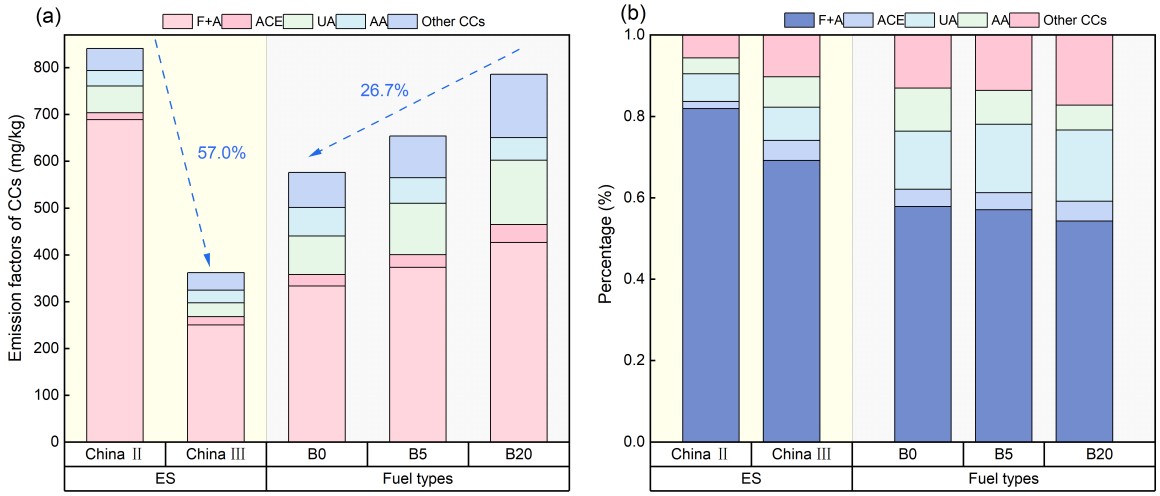

Figure 5: Differences in CCs emission characteristics(a)and composition(b)of non-road agricultural machinery

Upon analyzing the composition of CCs, it is observed that formaldehyde, acetaldehyde, and acrolein are the most abundant species in the emissions of CCs from AMs(He et al., 2009), accounting for 71.8% to 73.9% of the total emissions (Figure 5b). However, as the proportion of biodiesel blending increases, the proportion of formaldehyde and acetaldehyde decreases by 6.2%, while the combined proportion of heavier compounds such as acetone, unsaturated aldehydes, and other aldehyde and ketone substances increases by 20.0%. This corresponds exactly to the 20% addition ratio of biodiesel (v/v), indicating that the additional heavier CCs are entirely derived from the newly added biodiesel. Biodiesel, primarily composed of long-chain esters, is more likely to generate larger aldehyde and ketone compounds upon pyrolysis. The proportion of aromatic aldehydes is found to slightly decreased (by 4.5%) with the increase in biodiesel ratio, possibly because the addition of biodiesel reduces the content of aromatic hydrocarbons in the fuel, which are precursors for the formation of aromatic aldehydes. Overall, the blending of biodiesel significantly affects the composition of CCs, particularly by increasing the generation of heavier aldehyde and ketone compounds while reducing the emissions of small molecules such as formaldehyde and acetaldehyde.

**3.4 Emission Characteristics and Oxidative Potential Assessment of CCs from Different Combustion Sources**

**3.4.1 Emission Characteristics and Composition Comparison of CCs from Different Combustion Source**

Significant differences are observed in the composition of CCs between residential solid fuel sources and mobile sources (on-road and non-road vehicles). The combustion temperature is a key factor affecting the formation of small-molecule CCs, including formaldehyde and acetaldehyde. As depicted in Fig. 6, the proportion of formaldehyde and acetaldehyde in residential solid fuel sources (69.2%) is typically higher than that in mobile sources (57.1%), which can be attributed to differences in combustion temperature. The combustion temperature from residential solid fuel sources (such as straw, wood,

and residential coal), typically ranges lower (500– 800℃), in contrast to the higher temperature (1800–2000℃) of mobile sources. Under high-temperature conditions, formaldehyde and acetaldehyde, as highly reactive substances, are prone to polymerization and cyclization reactions, leading to a reduction of 12.1% in their proportion in emissions. It is evident that the formation of formaldehyde and acetaldehyde is more significantly influenced by combustion temperature. For the CCs emitted from DVs, the proportion of formaldehyde and acetaldehyde (44.8%) is lower than that from GVs (60.1%), which is due to the ignition method of DVs that allows for complete combustion of fuel, resulting in the full oxidation of aldehyde and ketone compounds (Saliba et al., 2017; Guo et al., 2023). In contrast, AMs, despite using the same fuel, exhibits a higher proportion of formaldehyde and acetaldehyde (57.9%) compared to DVs (44.8%). This difference may be attributed to the stricter emission standard and exhaust treatment technologies adopted by on-road vehicles.

The formation of acetone and aromatic aldehydes is significantly influenced by different fuel types. For instance, the proportion of acetone and aromatic aldehydes in RCC (20.9%) is higher than that in BB (10.0%), and this difference is mainly related to the chemical composition of the fuel. Residential coal is noted to contain a higher concentration of aromatic compounds, providing abundant precursors for the formation of acetone and aromatic aldehydes(Liu et al., 2022). With increasing temperature, the proportions of acetone and aromatic aldehydes in BB and RCC increase from 6.3%, 22.5% to 26.0%, 29.6%, respectively. Under high-temperature conditions, small molecule substances are more likely to undergo polymerization and cyclization reactions, leading to generation of a greater number of aromatic hydrocarbon compounds (Zhao et al., 2019). Similarly, the proportion of acetone and aromatic aldehydes in emissions from mobile sources, especially DVs, is found to be the highest (25.7–42.3%), possibly due to higher content of aromatic hydrocarbons in diesel (29–34%), which oxidize at high temperature to form acetone and aromatic aldehydes. Unlike the aforementioned CCs species, combustion temperature and fuel type have an impact on the emissions of unsaturated aldehydes. For example, the emission characteristics of unsaturated aldehydes differ notably. Owning to the presence of unsaturated bonds, unsaturated aldehydes have higher reactivity and cannot exist stably under high-temperature conditions, leading to a decrease in their proportion from 8.6% to 2.5% under high-temperature conditions. However, in non-road sources, the proportion of unsaturated aldehydes is observed to increases by approximately 20.4% with the increase in the proportion of biodiesel blending. This phenomenon can be explained by the chemical characteristics of biodiesel. Studies have shown that glycerol in biodiesel dehydrates and condenses to form acrolein under high-temperature conditions (Abdullah, 2022; Corma et al., 2008), resulting in an increase in the proportion of unsaturated aldehyde substances with the increase in the proportion of biodiesel.

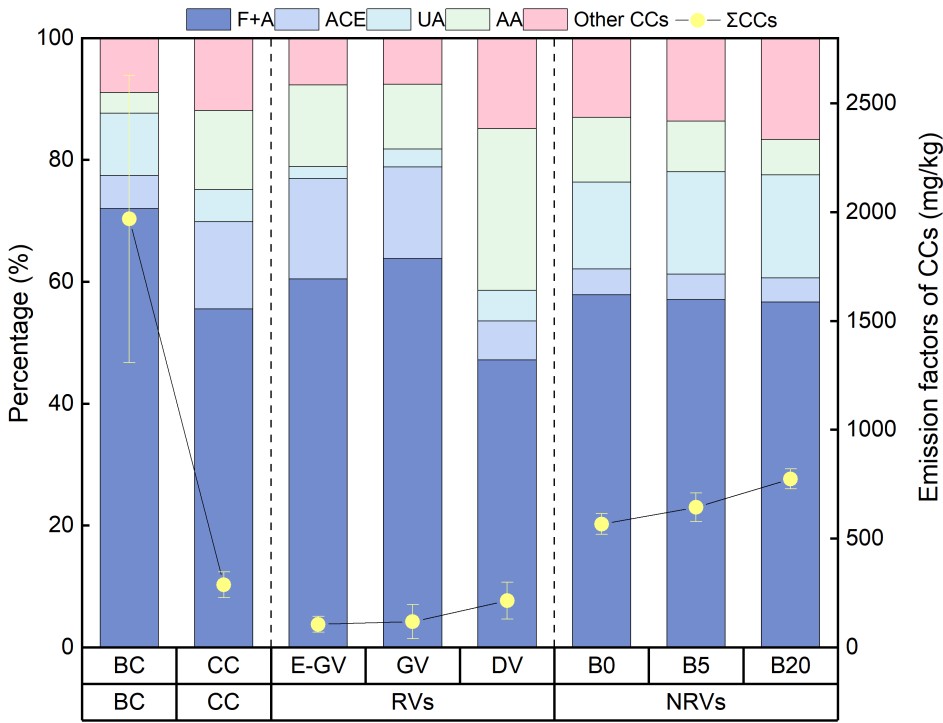

 **Figure 6: Differences in CCs emission characteristics and composition of four typical combustion sources; RV, on-road vehicle; NRV, non-road vehicle.**

### 3.4.2 Comparison of OFPs for CCs from Different Combustion Sources

In this study, the selection was made for calculation of the top ten CCs that have the greatest impact on the OFPs, which include formaldehyde, acetaldehyde, acetone, acrolein, propanal, crotonaldehyde, butanal, cyclohexanone, isopentanal, and pentanal. As illustrated in Fig. 7, significant differences are observed in the distribution of OFPs for the top ten CCs emanating from various combustion sources. The OFPs from BB sources (8.8 g O$_3$/kg-fuel) are found to be significantly higher than those from other sources ($p < 0.05$), being approximately 3.1 to 12.9 times higher than other sources, thereby indicating a substantial contribution of BB sources to the OFPs. The next highest OFPs are from non-road AMs sources (2.5 g O$_3$/kg-fuel), which are significantly higher than those from RCC sources (1.3 g O$_3$/kg-fuel) and on-road sources (0.4–1.1 g O$_3$/kg-fuel). Furthermore, it is noted that the species with high OFPs from all combustion sources are predominantly low-carbon CCs, such as formaldehyde and acetaldehyde, a finding that is consistent with other studies (Dong et al., 2014). This consistency further confirms that low-carbon CCs exhibit stronger reactivity and are present at higher concentrations in the atmosphere.

In on-road sources, the OFPs for E-GVs, GVs, and DVs are 0.4, 0.6, and 1.1 g O$_3$/kg-fuel, respectively. When compared to pure gasoline, it has been found that the use of ethanol-blended gasoline can lead to the reduction in OFPs by 31.0% (Yao et

al., 2011). This reduction is attributed to the fact that the CCs produced by E-GVs contain a higher proportion of acetaldehyde, which possesses a lower MIR value in comparison to formaldehyde. Consequently, while the use of ethanol-blended gasoline does not significantly diminish CCs emissions, it does contribute to a reduction in the reliance on fossil fuels and, to a certain extent, aids in diminishing the contribution of on-road sources to atmospheric oxidizing capacity. The OFPs obtained for GVs in this study are slightly higher than those reported by Cao et al. (2016) (0.5 g $O_3$/kg-fuel), while the OFPs for DVs is similar to the findings of Dong et al.(2014) (1.1 g $O_3$/kg-fuel) but slightly lower than that reported by Yao et al. (2015)(1.5 g $O_3$/kg-fuel). These differences may be related to the selection of MIR values, vehicle types, and driving distances (Wang et al., 2013), all of which can affect the composition of CCs and subsequently influence the OFPs values.

For AMs, the application of biodiesel also exerts a significant impact on OFPs. As shown in Fig. 7, the OFPs of B20 (3.5 g $O_3$/kg-fuel) is 1.4 times that of B0 (2.5 g $O_3$/kg-fuel), indicating that the use of biodiesel enhances the OFPs, thereby increasing atmospheric oxidizing capacity. In the emissions of CCs from AMs, formaldehyde, acetaldehyde, and acrolein are identified as the primary contributors to ozone formation, with formaldehyde being the most significant contributor, accounting for 56.3% to 57.9% of the total OFPs. With the increase in the proportion of biodiesel blending, the contribution of formaldehyde slightly decreases, while the contributions of acetaldehyde, propanal, and acrolein increase by 24.0%, possibly due to the presence of esters and glycerol in biodiesel. Acrolein is classified as a Group III carcinogen by the International Agency for Research on Cancer of the World Health Organization. Therefore, when employing biodiesel, it is necessary to consider the impact of newly generated toxic and harmful components in the exhaust on human health. The OFPs value obtained for AMs in this study is comparable to that reported by Yu et al. (2023) (2.2 g $O_3$/kg-fuel), and formaldehyde, acetaldehyde, and acrolein are also the main contributors to the OFPs of AMs(Yu et al., 2023).

Therefore, emissions from BB and AMs are significant sources of regional atmospheric oxidizing capacity and require attention. Appropriate control measures should be implemented, such as continuing to implement straw burning bans and vigorously promoting the use of energy-efficient stoves, which can reduce CCs emissions and further decrease ozone formation by 97.6%, thereby reducing atmospheric oxidizing capacity. For AMs, timely updates to emission standard (resulting in the 62.6% reduction in OFPs) can effectively decrease ozone formation, and efforts should be accelerated to explore new alternative fuels to reduce CCs emissions and lower atmospheric oxidizing capacity. Although biodiesel has good emission reduction effects on other conventional pollutants, it increases the proportion of some toxic components and atmospheric oxidizing capacity.

The OFPs reduction effects of the key emission control measures proposed in this study were evaluated, with the specific mitigation outcomes as follows.   For example, at a high combustion temperature of 800°C, the OFPs of BB and RCC are 0.8 (g $O_3$/kg-fuel) and 0.6 (g $O_3$/kg-fuel), respectively, indicating that increasing the combustion temperature can reduce ozone formation by 91.0% and 53.8%, respectively. Similarly, for both on-road and non-road sources, upgrading vehicle emission standard can significantly reduce ozone formation (46.8%~65.0%), with the most notable reduction effects observed for DV (63.6% reduction) and AMs (65.0% reduction), which are the sources with higher emission levels. In conclusion, the proposed measures in this study demonstrate significant emission reduction effects on atmospheric oxidizing capacity.

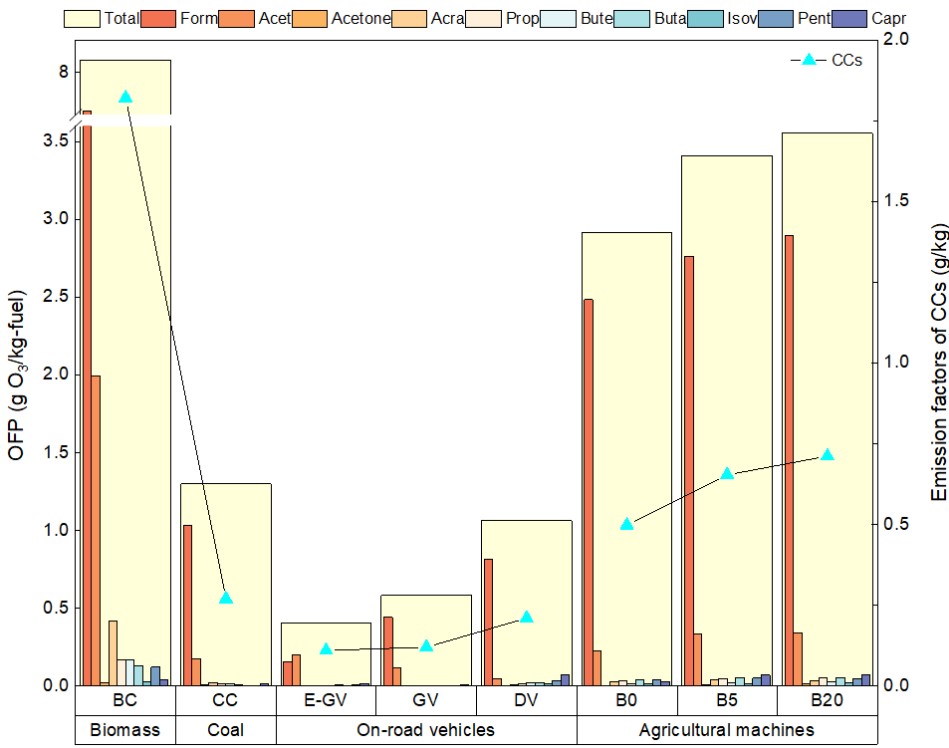

**Figure 7: Ozone formation potentials (OFPs) of the top ten carbonyl compounds emitted by different combustion sources.**

**4 Conclusions and Implication**

The emission characteristics of CCs from BB sources, RCC sources, on-road sources, and non-road AMs sources are found to exhibit significant differences. Specifically, the $EF_{CCs}$ from BB ($1968.2 \pm 661.2$ mg/kg) are an order of magnitude higher than those from on-road sources (117.8 to 214.1 mg/kg). Fuel type determines the composition of CCs: oxygen-rich BB sources primarily emit formaldehyde and acetaldehyde (80%), which is significantly higher than other combustion sources; whereas RCC sources, characterized by their aromatic structures, have a higher proportion of acetone and aromatic aldehydes (26.0%). In on-road sources, the addition of ethanol-blended gasoline significantly alters the ratio of formaldehyde to acetaldehyde, changing it from 5.6:1 to 1:1. Similarly, the use of biodiesel increases the production of unsaturated aldehydes by 20.4%, which corresponds exactly to the 20% (v/v) addition of biodiesel.

For residential solid fuel sources, the combustion temperature is identified as the key factor influencing the formation of CCs. Specifically, improving stove technology from 500℃ to 800℃ (high temperature) can reduce CCs emissions by 94.6%. For on-road sources and non-road AMs sources, the raising of emission standard has been found to have the significant reduction effect on high-emitting vehicles (61.3%). Therefore, when formulating emission reduction measures, priority should be given

to updating emission standard, followed by the use of new alternative fuels (such as ethanol-blended gasoline), which can also effectively reduce CCs emissions. However, it is important to note that after implementing corresponding emission reduction measures, the proportion of some toxic compounds may increase significantly. For example, although high temperature can enhance combustion efficiency and reduce the formation of CCs, close attention must be paid to acetone and aromatic
aldehydes. High temperature can cause small molecule substances to polymerize and cyclize, thereby promoting the formation of acetone and aromatic aldehydes, which also applies to liquid fuels. Additionally, new alternative fuels can change the composition of CCs: the use of ethanol-blended gasoline provides a large number of precursors for the formation of acetaldehyde, thereby increasing its proportion; on the other hand, the use of biodiesel promotes the formation of unsaturated aldehydes such as acrolein. Therefore, while considering combustion efficiency, attention should also be paid to the toxicity
and its impact on the environment and health.

Based on the calculation of OFPs using MIR values, it has been found that BB sources (8.8 g $O_3$/kg-fuel) and non-road AMs sources (3.5 g $O_3$/kg-fuel) contribute the most to the regional atmospheric OFPs, indicating that BB and AMs are significant sources of atmospheric oxidizing capacity. Moreover, for AMs, acrolein contributes significantly more to OFPs than other emission sources, thus highlighting the need for attention to BB sources and AMs sources. Targeted control measures should
be developed based on the composition of different combustion sources. At the same time, with the continuous growth in the number of on-road vehicles, the contribution of on-road sources emissions to atmospheric oxidizing capacity in the future should not be underestimated.

**Data availability**

All data of this study can be obtained by contact Yanli Feng(fengyanli@shu.edu.cn)

**Supplement**

Additional information can be found in the supporting information. Table S1 provides the uncertainty ranges for OFPs from different combustion sources. Table S2 provides details on the volatile content values of six raw coals. Table S3-S5 provide the basic information of on-road sources and non-road sources sampling vehicles, respectively. Table S6 provides details on the values of MIR. Text S1 provides the uncertainty analysis for OFP calculations. Text S2 provides the quality assurance/
quality control (QA/QC).

## Author contributions

LYJ, FYL and CYJ designed this study. LYJ, PY and JMJ carried them out and completed the measurement of carbonyl compounds of all samples. LYJ wrote this manuscript with contributions from all co-authors. FYL, CYJ and CT revised this manuscript.

## Competing interests

The authors declare that they have no conflict of interest.

## Acknowledgement

This work was supported by the National Natural Science Foundation of China (Nos. 42177086,42477095, 42407133), National Disease Control and Prevention Administration Talent Training Project for Public Health (2023), and the key projects in the three-year plan of Shanghai municipal public health system (2023-2025) (GWVI-4).

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
