# Peer review of "Carbonyl compounds from typical combustion sources: emission characteristics, influencing factors, and their contribution to ozone formation"

_EGUsphere, 2025_

## Author Comment (AC3)

Uncertainty analysis:

| Combustion sources | Max | Min |
| --- | --- | --- |
| BB | 21.5% | -3.1% |
| RCC | 13.8% | -10.9% |
| E-GV | 3.66% | -2.68% |
| GV | 3.84% | -2.31% |
| DV | 8.38% | -6.63% |
| AM | 10.27% | -4.52% |

The schematic diagram of the sampling system:

---

## Author Response (AR2)

**Response to the Reviewers' comments**

**Dear Editor:**

First, we would like to thank you and the anonymous reviewers for the constructive comments and suggestions. We have carefully addressed all comments and revised our manuscript accordingly. We think these opinions and suggestions can help to improve the quality of the manuscript. We have submitted three documents, including:

1. A clean copy of the revised manuscript, named "Clean Manuscript."

2. A copy of the revised manuscript with all changes highlighted, named "Revised Manuscript."

3. A detailed, point-by-point list of our replies to the comments of the reviewers, named "Response to comments."

Please find below our detailed response to the comments point-by-point. In addition, the manuscript has been improved according to the manuscript formatting request. Hope these will make the manuscript more acceptable for publication in Atmospheric Chemistry and Physics.

Yours sincerely,

Yingjun Chen

**Reviewer(s)' Comments to Author:**

Our point-to-point responses are presented in the following. We hope that the revision would satisfactorily address the comments and concerns of the editors and reviews.

**Reviewer #1:**

This manuscript focused on carbonyl compounds emitted by four types of combustion sources, including biomass burning, residential coal combustion, on-road vehicles and non-road mobile machineries. The emission factors were carefully determined and their influencing factors were comprehensively discussed. I think this manuscript could be considered for publication in ACP, after minor revisions.

**Response:** Thanks for your positive comments. We have carefully revised this manuscript based on the reviewer's comments. We hope that the revisions and improvements would satisfactorily address the reviewer's concerns. Our point-to-point responses are as follows.

1. Major point: Section 2.1, softwood and hardwood should be clearly distinguished. Section 3.1, please clarify whether the emission characteristics were comparable between softwood and hardwood. If not, please give possible explanations.

**Response:** Thanks for your valuable comments. We sincerely apologize for the oversight in not clearly distinguishing between softwood and hardwood in the original manuscript. In the revised version, we have explicitly differentiated these two fuel types in Section 2.1. Additionally, in Section 3.1, we have added a detailed comparison of the emission characteristics between softwood and hardwood. Our analysis reveals that softwood (e.g., pine) tends to emit higher concentrations of carbonyl compounds, particularly small-molecule carbonyl compounds such as formaldehyde and acetaldehyde, which is likely attributed to differences in lignin content and combustion efficiency. This finding has been discussed in the revised manuscript, along with potential explanations for the observed variations.

A detailed comparative analysis of carbonyl compound emission profiles across different fuel types has been incorporated, with new findings presented in Section 3.1 (Lines 212-231).

2. Abstract, re-write the first sentence as "Fuel combustions are important primary sources of carbonyl compounds (CCs)……"

**Response:** Thanks for your valuable comments. As suggested, we have rephrased the first sentence of the Abstract to: "Fuel combustion is an important primary sources of carbonyl compounds (CCs) ……" This revision improves the clarity and precision of the statement.

See the revised version in the Abstract section (Line 18).

3. Abstract, line 30, suggest removing the "tend".

**Response:** Thanks for your valuable comments. The word "tend" has been removed as recommended. The revised sentence now reads: "High-temperature promotes small molecules like F+A to cyclize, supplying ample precursors for the formation of acetone and aromatic aldehydes."
See the revised version in the Abstract section (Line 30).

4. I noticed that "EFccs" appeared only once. Remove this abbreviation or use it to replace "EFs of CCs" throughout the manuscript.

**Response:** Thanks for your valuable comments. We have carefully reviewed the manuscript and replaced all instances of "EFs of CCs" with "$EF_{CCs}$" to ensure consistency. This change has been applied throughout the text.
All modifications concerning $EF_{CCs}$ have been highlighted in red throughout the annotated manuscript. Given the extensive nature of these revisions, a comprehensive enumeration has been omitted for conciseness.

5. Some abbreviations were repeatedly defined. For example, CCs was defined twice in Page 2; EFs was defined again in Page 5.

**Response:** Thanks for your valuable comments. We apologize for the redundancy in defining abbreviations. In the revised manuscript, we have ensured that each abbreviation (e.g., CCs, EFs) is defined only once upon its first appearance.
The definition of carbonyl compounds (CCs) appears exclusively in two locations in the revised manuscript: Line 18 of the Abstract section and Line 35 of the Introduction.

6. Grammar mistakes like "Fuel types is key factors…(Page 1)" and "Fuel types determines the composition of…(Page 15)" should be avoided.

**Response:** Thanks for your valuable comments. We sincerely apologize for the grammatical errors in the original manuscript. The following corrections have been made:
"Fuel types is key factors…" has been corrected to "Fuel type is a key factor…"

"Fuel types determines the composition of…" has been corrected to "Fuel type determines the composition of…"

All other instances of subject-verb disagreement have been carefully reviewed and corrected.

The revised content appears in two locations: (1) Line 18 of the Abstract section, and (2) Line 450 of the Conclusions and Future Perspectives section.

7. Equations in Page 5, suggest using hyphens, which could be clearly distinguished from the minus signs, for the subscripts (e.g., c-fuel).

**Response:** Thanks for your valuable comments. Thank you for pointing out the inconsistency in the use of hyphens and minus signs in the equations. We have revised the equations to use hyphens for subscripts (e.g., c-fuel) and minus signs for mathematical operations, ensuring clarity and consistency.

The revised content appears in Section 2.3, page 6.

8. Page 7. Suggest briefly explaining the "bell-shaped distribution theory".

**Response:** Thanks for your valuable comments. We have added a brief explanation of the "bell-shaped distribution theory" in the revised manuscript. Specifically, we describe it as a theoretical framework used to explain the relationship between the volatile content of residential coal and its emission characteristics, where emissions peak at intermediate volatile content and decrease at both very high and very low volatile content levels. This addition provides better context for readers unfamiliar with the concept.

**Reviewer #2:**

This manuscript presents a comprehensive and in-depth study on the emission characteristics and influencing factors of carbonyl compounds (CCs) from various typical combustion sources. It provides the latest data for emission inventories and offers a scientific basis for targeted emission reduction strategies. This manuscript will make a significant contribution to the control and management of air pollution.

**Response:** We sincerely appreciate your thorough and constructive review of our manuscript titled "Emission Characteristics and Influencing Factors of Carbonyl Compounds from Various Typical Combustion Sources." We are grateful for your recognition of the manuscript's potential contribution to air pollution control and management, as well as for your detailed suggestions, which have significantly improved the quality of our work. Below, we provide a point-by-point response to your comments and outline the revisions we have made accordingly.

1. The abstract should be revised to better articulate the scientific innovation and applied value of this study, specifically by explicitly stating its contributions to optimizing combustion systems, guiding emission control policies, and enhancing atmospheric chemistry modeling frameworks.

**Response:** Thanks for your valuable comments. We have revised the Abstract to better highlight the scientific innovation and applied value of this study. Specifically, we have explicitly stated its contributions to optimizing combustion systems, guiding emission control policies, and enhancing atmospheric chemistry modeling frameworks. The revised Abstract now reads:
"This study provides the latest data for emission inventories and offers a scientific basis for targeted emission reduction strategies. Its findings contribute to optimizing combustion systems, guiding emission control policies, and enhancing atmospheric chemistry modeling frameworks."
The newly added content appears in the Abstract section (Lines 33-34).

2. The conclusion section should include specific estimates of the emission reduction effects of the proposed measures on atmospheric oxidizing capacity. For example, the impact of increasing combustion temperature and upgrading emission standards on ozone ($O_3$) formation should be quantified.

**Response:** Thanks for your valuable comments. As suggested, we have added specific estimates of the emission reduction effects of the proposed measures on atmospheric oxidizing capacity. For example, we have quantified the impact of increasing combustion temperature and upgrading emission standards on ozone ($O_3$) formation. The revised Conclusion now includes:
"At a high combustion temperature of 800°C, the ozone formation potentials (OFP) of BB and RCC

are 0.8 (g $O_3$/kg-fuel) and 0.6 (g $O_3$/kg-fuel), respectively, indicating that increasing the combustion temperature can reduce ozone formation by 91.0% and 53.8%, respectively. Similarly, for both on-road and non-road sources, upgrading vehicle emission standards can significantly reduce ozone formation (46.8%~65.0%), with the most notable reduction effects observed for DV (63.6% reduction) and AM (65.0% reduction), which are the sources with higher emission levels. In conclusion, the proposed measures in this study demonstrate significant emission reduction effects on atmospheric oxidizing capacity. ”

The additional content can be found in the 3.4.2 section (Lines 438 to 444, Page 17).

3. Spelling errors. In line 32 of the abstract, "allevite" should be corrected to "alleviate".

**Response:** Thanks for your valuable comments. We sincerely apologize for the spelling error in line 32 of the Abstract. The word "allevite" has been corrected to "alleviate."

The revised content appears in line 32.

4. Terms such as "$EF_{CCs}$" should be defined upon their first appearance, and consistent usage should be maintained throughout the manuscript. For example, "$EF_{CCs}$" and "EFs of CCs" should be standardized. Additionally, some abbreviations need to be redefined to avoid confusion with commonly used abbreviations, such as AA and OA.

**Response:** Thanks for your valuable comments. We have ensured that "$EF_{CCs}$" is defined upon its first appearance and used consistently throughout the manuscript. Additionally, In the fields of chemistry and environmental science, "AA" is a commonly used abbreviation for aromatic aldehydes. However, to avoid confusion with other terms, its definition has been explicitly stated upon its first use in this manuscript. Therefore, "AA" has been retained as the abbreviation for aromatic aldehydes in this article. As for "OA", which represents "other aldehydes and ketones" in this study, it is indeed prone to confusion with "organic aerosols" (OA). To address this, the term "other aldehydes and ketones" has been replaced with "Other CCs" throughout the manuscript.

All modifications concerning $EF_{CCs}$ have been highlighted in red throughout the annotated manuscript. Given the extensive nature of these revisions, a comprehensive enumeration has been

omitted for conciseness. Furthermore, all instances of 'OA' have been systematically replaced with 'Other CCs' throughout the manuscript, with corresponding revisions made to all figures and tables to maintain consistency.

5. Some sentences are lengthy and complex. Simplifying sentence structures can improve readability. For instance, the second paragraph of the introduction (line 49) could be broken into multiple sentences, and it's necessary to add the following reference in revised manuscript.

Emission of Intermediate Volatile Organic Compounds from Animal Dung and Coal Combustion and Its Contribution to Secondary Organic Aerosol Formation in Qinghai-Tibet Plateau, China. *Environmental Science & Technology* 2024 58 (25), 11118-11127.

**Response:** Thanks for your valuable comments. We have simplified lengthy and complex sentences to improve readability. For instance, the second paragraph of the Introduction (line 49) has been broken into multiple sentences. The revised Abstract now reads:

"Cheng et al. (2022) investigated the emission characteristics of CCs from BB, such as emission factors (EFs) and chemical composition, using a tube-furnace. They found that combustion conditions, including temperature and oxygen concentration, as well as fuel characteristics like composition, significantly affect the EFs and composition of CCs."

We have also added the suggested reference in the second paragraph of the introduction:

"Emission of Intermediate Volatile Organic Compounds from Animal Dung and Coal Combustion and Its Contribution to Secondary Organic Aerosol Formation in Qinghai-Tibet Plateau, China. *Environmental Science & Technology* 2024 58 (25), 11118-11127."

The revised content appears on page 2, line 51. The newly added references are presented on page 2, lines 56-58.

6. Line 105: For biomass samples, the selection of 2g is difficult. Besides, the combustion process is very fast. How can the author ensure the accuracy of sampling throughout the entire combustion process? the following are some references related to experiment process that can be cited:

Examination of long-time aging process on volatile organic compounds emitted from solid fuel combustion in a rural area of China. *Chemosphere* 333 (2023) 138957.

**Response:** Thanks for your valuable comments. We acknowledge the concern regarding the selection of 2g biomass samples and the rapid combustion process. To ensure sampling accuracy, we have implemented a continuous sampling protocol and validated it through repeated experiments. We have also cited the following reference to support our methodology:

"Examination of long-time aging process on volatile organic compounds emitted from solid fuel combustion in a rural area of China. *Chemosphere* 333 (2023) 138957."

The newly added references are presented on page 4, lines 126.

7. Ensure that the font size in all figures and tables is consistent to present a more organized and visually appealing layout. For example, the legend font size in Figure 1a is slightly larger than in the other three subfigures.

**Response:** Thanks for your valuable comments. We have ensured that the font size in all figures and tables is consistent. For example, the legend font size in Figure 1a has been adjusted to match the other subfigures, presenting a more organized and visually appealing layout.

8. The manuscript should include (1) uncertainty quantification analyses of ozone formation potential across diverse CCS emission sources, and (2) a schematic diagram of the sampling system in Section 2.1 to enhance methodological transparency.

**Response:** Thanks for your valuable comments. We have added (1) uncertainty quantification analyses of ozone formation potential across diverse CCs emission sources, and (2) a schematic diagram of the sampling system in Section 2.1 to enhance methodological transparency. The schematic diagram illustrates the sampling setup and process, providing readers with a clear understanding of the experimental design. Here are some of the new additions:

(1) Uncertainty quantification analyse:

| Combustion sources | Max | Min |
|---|---|---|
| BB | 21.5% | -3.1% |
| RCC | 13.8% | -10.9% |
| E-GV | 3.66% | -2.68% |
| GV | 3.84% | -2.31% |
| DV | 8.38% | -6.63% |
| AM | 10.27% | -4.52% |

The table presents the uncertainty ranges of ozone formation potential estimates for different combustion sources, expressed as maximum (Max) and minimum (Min) deviation percentages. It can be observed that the deviations are generally positively correlated with the emission factors of carbonyl compounds, meaning that the higher the emission factor, the greater the uncertainty in the calculated ozone formation potential for the combustion source. Among these, BB, RCC, and RCC are identified as high-uncertainty combustion sources, indicating that their emission characteristics are complex and may significantly contribute to ozone formation. It is recommended to conduct more precise measurements and modeling of the emission factors for these combustion sources to reduce uncertainty.

(2) Schematic diagram of the sampling system:

[Figure]

The newly added schematic diagram of the sampling system appears in Section 2.2 (page 5), while the additional uncertainty analysis is presented in Supplement Text S1(page 2).

9. The figures should be replaced with higher resolution versions to ensure graphical clarity essential for proper interpretation of the experimental data.

**Response:** Thanks for your valuable comments. All figures have been replaced with higher resolution versions to ensure graphical clarity essential for proper interpretation of the experimental data.

**Reviewer #3:**

This manuscript conducts an in-depth investigation into the emission characteristics and influencing factors of carbonyl compounds (CCs) from four typical combustion sources, based on laboratory simulations and road measurements. The findings fill the data gap regarding the emission characteristics of CCs from different combustion sources, providing significant scientific and innovative value for atmospheric pollution control and policy formulation.

**Response:** Thanks for your valuable comments.We sincerely thank the reviewer for their thorough and constructive feedback on our manuscript. We greatly appreciate the time and effort dedicated to evaluating our work and providing valuable suggestions for improvement.

1. The experimental design mentions a more detailed classification of fuel types, but the results and discussion section does not address these finer distinctions, such as whether there are differences in CCs emissions between southern and northern straw burning, or between softwood and hardwood.

**Response:** Thanks for your valuable comments. We appreciate the reviewer's observation regarding the finer distinctions in fuel types, such as differences in carbonyl compounds (CCs) emissions between southern and northern straw burning, as well as between softwood and hardwood. In response, we have expanded the Results and Discussion section to include a more detailed analysis of these distinctions. Specifically, we have added a subsection comparing the emission characteristics of CCs from southern and northern straw burning, as well as softwood versus hardwood combustion. These additions are supported by additional data and references to relevant studies. Below is a brief description of the newly added content:

The carbonyl compounds generated from the combustion of southern straw (rice straw) are higher than those from northern straw. The emission factor of CC from rice straw combustion (3865.6 ±

558 mg/kg) is significantly higher than that of corn (2829.8 ± 1771.8 mg/kg) and wheat (1772.6 ± 847.2 mg/kg), indicating that the carbonyl compounds generated from southern straw combustion are 1.4-2.2 times higher than those from northern straw combustion. This phenomenon may be attributed to differences in the content of biomass components (cellulose, hemicellulose, and lignin) and combustion efficiency. Cheng et al. (2022) found that among the three biomass components, cellulose combustion generates the highest amount of CCs, followed by hemicellulose, with lignin producing the least. Meanwhile, Zhao et al. (2019) discovered that the holocellulose content (cellulose + hemicellulose: ~56.3%) of rice straw is higher than that of corn and wheat. Additionally, in this study, the combustion efficiency of rice straw (90.9%) is slightly lower than that of corn and wheat (92.0%-92.8%).

The carbonyl compounds generated from the combustion of softwood (pine) are higher than those from hardwood. The emission factor of CCs from pine combustion (1415.3 ± 431.8 mg/kg) is significantly higher than that of poplar (1020.3 ± 249.1 mg/kg) and willow (905.5 ± 109.6 mg/kg), indicating that the carbonyl compounds generated from softwood combustion are 1.4-1.6 times higher than those from hardwood. This phenomenon may be due to differences in combustion efficiency. In this study, the combustion efficiency of pine (94.0%) is significantly lower than that of poplar and willow (95.6%-96.0%). Studies have shown that incomplete combustion of fuels is more likely to generate CCs.

A detailed comparative analysis of carbonyl compound emission profiles across different fuel types has been incorporated, with new findings presented in Section 3.1 (Lines 212-233).

2. While the discussion on residential solid fuel combustion is thorough, the analysis of the formation mechanisms of CCs from on-road sources, particularly for ethanol-blended gasoline and biodiesel, is relatively brief. It is recommended to include additional discussion on the formation mechanisms of CCs from ethanol-blended gasoline and biodiesel, along with relevant references.

**Response:** Thanks for your valuable comments. We agree with the reviewer that the discussion on the formation mechanisms of CCs from on-road sources, particularly for ethanol-blended gasoline and biodiesel, was relatively brief. To address this, we have expanded the discussion in Section 4.2 to include a more detailed analysis of the formation mechanisms of CCs from ethanol-blended

gasoline and biodiesel. We have also added relevant references to support this discussion, ensuring a more comprehensive understanding of the underlying processes.

For the ethanol gasoline section, an in-depth discussion on different emission standards has been added. It was found that the "acetaldehyde/formaldehyde" ratio in carbonyl compounds emitted by China VI emission standard ethanol gasoline vehicles is approximately 0.38, which is about 2.5 times that of gasoline vehicles under the same emission standard. This indicates that the use of ethanol gasoline significantly increases acetaldehyde emissions. In contrast, this ratio for China V ethanol gasoline vehicles is 1.80, suggesting that higher emission standards can effectively reduce carbonyl compound emissions while also decreasing acetaldehyde generation. The reason for this phenomenon may be that, as emission standards are continuously upgraded, exhaust treatment technologies are improved, and emission requirements become stricter, leading to further oxidation of harmful substances such as acetaldehyde into formaldehyde.

For the biodiesel section, two additional references have been included to enhance the understanding of the formation mechanisms:

Lin, Y.C., Hsu, K.H., and Chen, C.B.: Experimental investigation of the performance and emissions of a heavy-duty diesel engine fueled with waste cooking oil biodiesel/ultra-low sulfur diesel blends, *Energy*, 36, 241–248, https://doi.org/10.1016/j.

energy.2010.10.045, 2011.

Chien, S.M., Huang, Y.J., Chuang, S.C., and Yang, H.H.: Effects of biodiesel blending on particulate and polycyclic aromatic hydrocarbon emissions in nano/ultrafine/fine/ coarse ranges from diesel engine, *Aerosol Air Qual Res*., 9, 18–31, https://doi.org /10.4209/aaqr.2008.

09.0040, 2009.

The expanded discussion on ethanol gasoline appears on page 12 (lines 285-291), while the additional references for biodiesel are presented on page 14 (line 343).

3. Grammatical errors. There are several instances of subject-verb disagreement in the text, for example, in line 24 of the abstract.

**Response:** Thanks for your valuable comments. We sincerely apologize for the grammatical errors in the manuscript, particularly the subject-verb disagreement in line 24 of the abstract. The revised

Abstract now reads:

"Fuel type is a key factor affecting the CCs components."

We have conducted a thorough proofreading of the entire manuscript to correct grammatical errors, improve sentence structure, and ensure consistency in language usage. All instances of subject-verb disagreement have been resolved.

The revised content appears in Line 24 of the Abstract section.

4. The terms $EF_{CCs}$ and EFs of CCs convey the same meaning. It is recommended to use $EF_{CCs}$ throughout the manuscript to replace "EFs of CCs".

**Response:** Thanks for your valuable comments. We thank the reviewer for pointing out the inconsistency in the use of $EF_{CCs}$ and EFs of CCs. As suggested, we have replaced all instances of "EFs of CCs" with "$EF_{CCs}$" throughout the manuscript to ensure consistency and clarity.

All modifications concerning $EF_{CCs}$ have been highlighted in red throughout the annotated manuscript. Given the extensive nature of these revisions, a comprehensive enumeration has been omitted for conciseness.

5. The first occurrence of an abbreviation in the main text should be clearly annotated with its full name. For example, "ES" appears for the first time in line 74 of the abstract without its full name being provided.

**Response:** Thanks for your valuable comments. We apologize for the oversight in not providing the full name for the abbreviation "ES" upon its first occurrence in line 74 of the abstract. We have now clearly annotated all abbreviations with their full names upon their first appearance in the main text.

"ES" is now defined as "emission standard" in line 74.

6. The font format in the formulas is inconsistent. It is recommended to unify the font format throughout.

**Response:** Thanks for your valuable comments. We acknowledge the inconsistency in the font

format used in the formulas. To address this, we have standardized the font format for all formulas throughout the manuscript, ensuring a consistent and professional presentation.

**Reviewer #4:**

1. There are far too many acronyms throughout the paper. The abstract is difficult to read with so many acronyms. There is no need to use acronyms for terms such as moisture content, residential solid fuel combustion etc.

Response: We sincerely appreciate the reviewer's constructive feedback on improving the manuscript's readability. We fully agree that excessive acronyms can hinder comprehension, particularly in critical sections like the abstract. In response, we have implemented the following revisions:

1). Abstract: Elimination of Non-Essential Acronyms

Restored " formaldehyde and acetaldehyde " in the abstract (2 instances revised), removing the abbreviated form "F+A";

Replaced "$EF_{CCs}$" with the full term "emission factor of CCs" in the abstract (2 instances revised);

Replaced "OFPs" with the full term "ozone formation potentials" in the abstract (2 instances revised);

Replaced "AMs" with the full term "agricultural machineries" in the abstract (3 instances revised).

The abstract maintains only two frequently used acronyms (CCs and BB), both properly defined upon their first occurrence.

2). Full-text Optimization

In the manuscript text, we have systematically replaced all low-frequency abbreviations with their full terms:

single-occurrence terms including $O_3$, SOA, and LDGVs were replaced with "ozone," "secondary organic aerosols," and "light-duty gasoline vehicles" respectively;

PEMs (2 instances revised) was substituted with "portable emission measurement system";

SC and WC (each 6 instances revised) were changed to "straw combustion" and "wood combustion" throughout;

MC was uniformly expressed as "moisture content," and RSFC was contextually replaced with either "BB and RCC" or "residential solid fuel combustion" depending on the specific discussion context.

2. Far too many significant figures in abstract, in Table 2 and in the conclusions.

Response: We sincerely appreciate the reviewer's insightful comments on improving data presentation. In response to the concern regarding excessive significant figures in the abstract, Table 2, and conclusions, we have implemented a systematic revision to enhance clarity while maintaining scientific accuracy. The key modifications include:

1). In both the abstract and conclusions, non-critical numerical values that do not affect the core conclusions were removed. Additionally, redundant or overlapping data points were consolidated.

2). To improve interpretability, Table 2 has been replaced with Figure 2, a bar chart that: Eliminates numerical clutter while preserving key trends. And the original tabular data is retained in Supplementary Table S7 for full transparency.

All changes have been highlighted in red in the revised manuscript for easy reference. We believe these adjustments significantly enhance readability without compromising scientific rigor.

3. Abstract: "Fuel combustion" should not be plural.

Response: We sincerely appreciate the reviewer's keen attention to linguistic accuracy. In response to the comment regarding the plural form of "fuel combustion," we have implemented the following corrections:

1). Revised "fuel combustions" to "fuel combustion" in line 18. The revised sentence now reads: " Fuel combustion is an important primary source of carbonyl compounds (CCs)..."

2). Systematically revised all instances throughout the manuscript to maintain singular form consistency. We are grateful for this suggestion, which has undoubtedly improved the manuscript's linguistic quality.

4. Line 61: awkward language "maturity of RCC"

Response: We sincerely appreciate the reviewer's valuable suggestion regarding terminology precision. In response to the comment about the phrasing "maturity of RCC", we have implemented the following improvements: We have revised the expression " maturity of RCC " to the more academically appropriate " maturity of residential coal ". The revised text is in line 60. Additionally, we have conducted a comprehensive review of all similar expressions in the manuscript. We believe

these modifications have significantly improved the academic rigor and clarity of our manuscript.

5. Line 63: awkward language "an urgent need to reveal".

Response: We sincerely appreciate the reviewer's astute observation regarding the phrasing in our manuscript. In response to your valuable comment, we have carefully refined the expression "an urgent need to reveal" to "it is crucial to investigate" for enhanced academic precision. This revision: This adjustment maintains the intended emphasis on research urgency while improving the linguistic fluency. The modified phrasing better conveys the significance of the research objective, with specific changes visible in line 62 in the revised manuscript.We believe this modification significantly strengthens the scholarly tone of our manuscript while preserving the original scientific intent.

6. Lines 63 and 95: $V_{daf}$ should include subscripts

Response: We are deeply grateful for the reviewer's exceptional attention to typographical precision. Regarding the subscript notation for "$V_{daf}$", we have implemented a rigorous verification and standardization process: we have conducted manual visual inspection of all 13 instances across the manuscript. And we confirmed proper subscript rendering in revised PDF and Word formats. We sincerely appreciate this opportunity to enhance our manuscript's technical accuracy.

7. Line 108: remove "the"

Response: We sincerely appreciate your attention to grammatical precision. The requested revision has been implemented as follows:

Original sentence (Line 108):

"...six types of coal with the different $V_{daf}$ (LL coal, GJ coal, DT coal, SH coal, NM coal, and PX coal)."

Revised version:

"...six types of coal with different $V_{daf}$ (LL coal, GJ coal, DT coal, SH coal, NM coal, and PX coal)."

We have screened and revised similar expressions throughout the text. Thank you for enhancing the linguistic quality of this work.

8. do these represent smoldering and flaming conditions?

Response: We appreciate the reviewer's insightful observation regarding the interpretation of the temperature settings (500°C and 800°C) in this study. To clarify, these temperature thresholds were primarily selected to highlight differences in stove combustion technology. Notably, the experimental conditions of 500 °C and 800 °C may exhibit an indirect correlation with smoldering and flaming combustion regimes, as indicated by metrics such as Modified Combustion Efficiency (MCE). Studies have demonstrated that an MCE below 0.9 indicates that over 50% of the combustion occurs in a smoldering phase, signifying a predominantly smoldering-dominated regime. Conversely, an MCE exceeding 0.9 suggests that flaming combustion accounts for more than half of the process. In this experiment, under the 500°C condition, the MCE values of the three tested fuels ranged from 0.65 to 0.90, whereas at 800°C, the MCE values varied between 0.91 and 0.99. These results demonstrate that the temperature settings in this study effectively characterize the emission profiles of fuels under both smoldering and flaming combustion conditions.

9. Line 150: modified combustion efficiency (MCE) is often used in the literature. Why not use MCE? (e.g. VOCs are missing from the carbon balance from this equation.)

Response: We sincerely apologize for the ambiguity in the original manuscript regarding the terminology related to combustion efficiency. To clarify, the Incomplete Combustion Factor (K) mentioned in the derivation of emission factors serves exclusively as a mathematical parameter for emission factor derivation. This coefficient is distinct from the widely recognized Modified Combustion Efficiency (MCE), which remains the primary metric for evaluating combustion efficiency in this study. The role of K is analogous to the Products of Incomplete Combustion (PIC) parameter introduced by Shen et al.(Shen et al., 2010), which was conceptualized to partition carbon fractions rather than to quantify combustion efficiency.

All combustion efficiency values discussed in the Results section were rigorously derived using the standardized MCE framework. The confusion likely arose from our failure to explicitly define the MCE calculation methodology in the original text (Lines 210–211). We acknowledge this oversight and have now added a detailed description of the MCE computational procedure in Section 2.3, including the formula:

$$MCE = \frac{\Delta CO_2}{(\Delta CO_2 + \Delta CO)}$$

where $CO_2$ and $CO$ are the concentrations of $CO_2$ and $CO$ emitted from the fuel combustion.

We deeply regret any inconvenience caused by the lack of clarity in the original manuscript and greatly appreciate the reviewer's meticulous attention to methodological transparency. Further revisions will be made promptly if additional specifications are required.

Li, Q., Jiang, J., Cai, S., Zhou, W., Wang, S., Duan, L., and Hao, J.: Gaseous ammonia emissions from coal and biomass combustion in household stoves with different combustion efficiencies, Environ. Sci., 2016.

McMeeking, G. R., Kreidenweis, S. M., Baker, S., Carrico, C. M., Chow, J. C., Collett, J. L., Hao, W. M., Holden, A. S., Kirchstetter, T. W., Malm, W. C., Moosmüller, H., Sullivan, A. P., and Wold, C. E.: Emissions of trace gases and aerosols during the open combustion of biomass in the laboratory, J. Geophys. Res.: Atmos., 114, 2009JD011836, https://doi.org/10.1029/2009JD011836, 2009.

Shen, G., Yang, Y., Wang, W., Tao, S., Zhu, C., Min, Y., Xue, M., Ding, J., Wang, B., Wang, R., Shen, H., Li, W., Wang, X., and Russell, A. G.: Emission factors of particulate matter and elemental carbon for crop residues and coals burned in typical household stoves in China, Environ. Sci. Technol., 44, 7157–7162, https://doi.org/10.1021/es101313y, 2010.

10. Section 2.5 seems to be more about the uncertainty in OFP calculations, not, as the title suggests, OFP itself.

Response: We are grateful for the reviewer's constructive suggestion to enhance the manuscript's focus. In addressing the comments on ozone formation potential (OFP) uncertainty analysis, we have: (1) relocated the detailed methodology from Section 2.5 to Supplement Text S1 to streamline the main text, and (2) added a clear cross-reference in Line 174: "Complete OFP uncertainty analysis appears in Supplement Text S1." These modifications maintain full methodological transparency while improving the paper's conceptual flow, and we sincerely appreciate this opportunity to strengthen our work.

11. Statistical method details should be mentioned in the method section. e.g. what is the type of t-test used?

Response: We sincerely thank the reviewer for their valuable suggestion. In response, we have added a new subsection on significance testing (Section 2.5) to explicitly clarify the types of t-tests employed in this study. The revised text reads as follows:

2.5 Statistical Significance Testing

The t-test is a widely utilized statistical method for comparing mean differences between samples. In this study, data analysis was performed using SPSS 26.0. To evaluate the significance of mean differences between the two groups, the normality of the data was first assessed via the Kolmogorov-Smirnov test. Upon confirmation of normality, the homogeneity of variances was subsequently examined using Levene's test. If variance homogeneity was satisfied, an independent samples t-test was applied for intergroup comparisons; otherwise, Welch's corrected t-test was employed. Statistical significance was determined at a 95% confidence interval (CI), with a two-tailed significance threshold set at $\alpha = 0.05$. Results were reported as t-values, degrees of freedom (df), and $p$-values. A statistically significant difference was defined as $p<0.05$. The analytical procedure strictly adhered to the classical t-test assumptions (independence, normality, and homogeneity of variance) to ensure methodological rigor.

This revision ensures methodological transparency and addresses the reviewer's concerns regarding the rigor of our statistical framework. Should further clarification be required, we are prepared to provide additional details.

12. Line 209: "within the fuel" is unclear. Clarify whether it is "for the same fuel", or physically inside the fuel.

Response: We sincerely appreciate your critique regarding the ambiguous phrasing "within the fuel." The text has been revised to clarify the source of variability as follows:

1). Original sentence:

Additionally, the large error bars in each type of fuel indicate variability in emissions within the fuel, which may be related to combustion conditions.

2). Revised version:

Additionally, significant error bars among replicate combustion tests of the same fuel suggest that the emission characteristics of carbonyl compounds may be related to combustion conditions.

We deeply value your guidance in enhancing methodological clarity.

13. Line 224-225: what is this combustion efficiency? 1 - K? As suggested earlier, MCE is more commonly used.

Response: We sincerely appreciate the reviewer's valuable comments regarding combustion efficiency clarification. To address this concern, we have: (1) explicitly stated that all combustion efficiency values were calculated using the standardized Modified Combustion Efficiency (MCE) metric, (2) emphasized its wide acceptance in biomass combustion research, and (3) implemented comprehensive textual revisions throughout the manuscript to ensure methodological transparency. These modifications have been highlighted in the revised version for easy reference.

In response to your critique, we have implemented comprehensive revisions as follows: we acknowledge this oversight and have now added a detailed description of the MCE computational procedure in Section 2.3:

The modified combustion efficiency (MCE) is a widely accepted evaluation parameter in biomass combustion research(Li et al., 2016; McMeeking et al., 2009; Shen et al., 2010), which can be used to represent the combustion condition:

$$MCE = \frac{\Delta CO_2}{(\Delta CO_2 + \Delta CO)}$$

where $CO_2$ and $CO$ are the concentrations of $CO_2$ and $CO$ emitted from the fuel combustion.

McMeeking, G. R., Kreidenweis, S. M., Baker, S., Carrico, C. M., Chow, J. C., Collett, J. L., Hao, W. M., Holden, A. S., Kirchstetter, T. W., Malm, W. C., Moosmüller, H., Sullivan, A. P., and Wold, C. E.: Emissions of trace gases and aerosols during the open combustion of biomass in the laboratory, *J. Geophys. Res.: Atmos.*, 114, 2009JD011836, https://doi.org/10.1029/2009JD011836, 2009.

Shen, G., Yang, Y., Wang, W., Tao, S., Zhu, C., Min, Y., Xue, M., Ding, J., Wang, B., Wang, R., Shen, H., Li, W., Wang, X., and Russell, A. G.: Emission factors of particulate matter and elemental carbon for crop residues and coals burned in typical household stoves in China, *Environ. Sci. Technol.*, 44, 7157–7162, https://doi.org/10.1021/es101313y, 2010.

Li, Q., Jiang, J., Cai, S., Zhou, W., Wang, S., Duan, L., and Hao, J.: Gaseous ammonia emissions from coal and biomass combustion in household stoves with different combustion efficiencies, *Environ. Sci.*, 2016.

14. Line 289: "ethanol gasoline utilization" is awkward. Rephrase

Response: We are grateful for the reviewer's insightful suggestions regarding terminology precision. In response, we have conducted a systematic revision of the term "utilization" throughout the

manuscript, replacing it with "application" where appropriate. This modification is based on: (1) a comprehensive literature review confirming that "application" better reflects the functional deployment of fuels in specific technological contexts (e.g., emission control, combustion optimization), and (2) its closer alignment with our study's focus on engineered fuel usage for performance-driven scenarios. All revisions have been carefully implemented and highlighted in the revised manuscript.

Specific Revisions

Line 284:

"The ethanol gasoline utilization..."

Revised to:

"The ethanol gasoline application..."

Line 416:

"the utilization of biodiesel..."

Revised to:

"the application of biodiesel..."

We sincerely appreciate your guidance on terminology precision.

15. Line 290: "emission standard is continuously upgraded". awkward. rephrase

Response: We sincerely appreciate your critique of the phrasing "emission standard is continuously upgraded." The revised text now aligns with regulatory terminology and academic precision:

1). Original sentence (Introduction section):

Line 71-72: " the continuous updating of emission standard..."

Line 287: "The emission standard is continuously upgraded..."

2). Revised version:

"the progressive tightening of emission standards..."

"With the progressive tightening of emission standards..."

3). Rationale for Revisions: "Progressive tightening" directly cites U.S. EPA policy documents, precisely conveying the phased stringency of emission regulations, replacing the ambiguous "continuously upgraded."

16. Line 308: Straight chain alkanes are not very prevalent in gasoline (it reduces its octane number). Stating that gasoline fuel have lower carbon numbers is probably sufficient.

Response: We sincerely appreciate your critique regarding the description of gasoline composition. The text has been systematically revised to align with the chemical realities of fuel formulation:

Original text in Lines 305-306:

" Gasoline, in contrast, primarily consists of lower carbon straight-chain alkanes (C4-C12) and cycloalkanes..."

Revised to:

" Gasoline, in contrast, primarily consists of lower carbon hydrocarbons (C4-C12)..."

We deeply value your guidance in enhancing the chemical precision of this work.

17. is biodiesel use strictly off-road

We sincerely appreciate the reviewer's inquiry. Below is a point-by-point clarification regarding the rationale behind our research design and its connection to biodiesel applications:

1). Application Context of Biodiesel in China

Existing studies indicate that biodiesel in China is predominantly utilized in the transportation sector, such as vehicle diesel (B5/B10 standard fuels) and marine fuel substitutes (refer to China Biodiesel Industry Development Report 2022). Recent academic research has focused on combustion optimization and emission characteristics in diesel vehicles/engines, aligning with China's decarbonization strategy for transportation sector emissions reduction.

2). Rationale for Selecting Non-Road Sources

The choice of agricultural machinery as the research subject is based on the following considerations:

a. Significant Emission Contribution: Non-road mobile sources (including agricultural machinery) account for 15%-20% of national NOx and PM emissions (China Mobile Source Environmental Management Annual Report 2021), making them critical targets for regional air pollution control.

b. Technological Representativeness: Current agricultural machinery predominantly employs diesel engines compliant with China III or lower emission standards, with a low penetration rate of exhaust after-treatment systems (<30%). These high-emission-intensity platforms, characterized by outdated technologies, provide a "high-sensitivity" experimental condition for elucidating the impact of biodiesel blending on carbonyl compound formation mechanisms.

c. Generalizability of Scientific Insights: Systematic study of such high-emission sources can reveal universal patterns of carbonyl compound formation influenced by biodiesel blending, establishing a unified theoretical framework for emission regulation in both road and non-road scenarios.

3). Scope of Research Conclusions

It is crucial to emphasize that our focus on agricultural machinery does not imply biodiesel is exclusively suitable for non-road applications. Instead, this study aims to identify generalized mechanisms through systematic analysis of emission characteristics in agricultural machinery.